# Structure and mechanism of cancer-associated N-acetylglucosaminyltransferase-V

Masamichi Nagae[1,2], Yasuhiko Kizuka[3,4], Emiko Mihara[5], Yu Kitago[5], Shinya Hanashima[6], Yukishige Ito [7], Junichi Takagi[5], Naoyuki Taniguchi[3] & Yoshiki Yamaguchi[1]

N-acetylglucosaminyltransferase-V (GnT-V) alters the structure of specific N-glycans by modifying α1-6-linked mannose with a β1-6-linked N-acetylglucosamine branch. β1-6 branch formation on cell surface receptors accelerates cancer metastasis, making GnT-V a promising target for drug development. However, the molecular basis of GnT-V's catalytic mechanism and substrate specificity are not fully understood. Here, we report crystal structures of human GnT-V luminal domain with a substrate analog. GnT-V luminal domain is composed of a GT-B fold and two accessory domains. Interestingly, two aromatic rings sandwich the α1-6 branch of the acceptor N-glycan and restrain the global conformation, partly explaining the fine branch specificity of GnT-V. In addition, interaction of the substrate N-glycoprotein with GnT-V likely contributes to protein-selective and site-specific glycan modification. In summary, the acceptor-GnT-V complex structure suggests a catalytic mechanism, explains the previously observed inhibition of GnT-V by branching enzyme GnT-III, and provides a basis for the rational design of drugs targeting N-glycan branching.

[1] Structural Glycobiology Team, Glycobiology Research Group, Global Research Cluster, RIKEN, 2-1 Hirosawa, Wako, Saitama 351-0198, Japan. [2] Graduate School of Pharmaceutical Sciences, The University of Tokyo, Hongo 7-3-1, Bunkyo-ku, Tokyo 113-0033, Japan. [3] Disease Glycomics Team, Systems Glycobiology Research Group, Global Research Cluster, RIKEN, 2-1 Hirosawa, Wako, Saitama 351-0198, Japan. [4] Center for Highly Advanced Integration of Nano and Life Sciences (G-CHAIN), Gifu University, 1-1 Yanagido, Gifu-City, Gifu 501-1193, Japan. [5] Institute for Protein Research, Osaka University, Suita, Osaka 565-0871, Japan. [6] Department of Chemistry, Osaka University, Machikaneyama, Toyonaka, Osaka 560-0043, Japan. [7] Synthetic Cellular Chemistry Laboratory, RIKEN, 2-1 Hirosawa, Wako, Saitama 351-0198, Japan. These authors contributed equally: Masamichi Nagae, Yasuhiko Kizuka. Correspondence and requests for materials should be addressed to M.N. (email: mnagae@mol.f.u-tokyo.ac.jp)

Glycosylation, the most common posttranslational modification, provides proteins with an extensive array of functional and structural variations, enabling homeostasis of complex multicellular biological systems despite a limited number of proteins[1,2]. Glycans on proteins have diverse physiological functions, including roles in early development, immunity, neural plasticity, etc[3,4]. There are many disease-associated alterations of glycans[5,6], and much interest currently in the rational design of glycan-based therapeutics. Disease-specific changes of glycans are indeed used clinically as reliable biomarkers particularly in the cancer context, such as AFP-L3, SLeX, and CA19-9[7,8]. In addition, numerous studies using glycan-engineered mice have demonstrated that genetic manipulation of glycans can improve pathological phenotypes in vivo[1,9], suggesting that the glycans on proteins are also potential drug targets for several diseases including cancer. Chemical shut down of a biosynthetic pathway of a disease-causing glycan would appear to

be a reasonable approach to developing novel drugs. As glycans are biosynthesized by concerted actions of glycosyltransferases in the endoplasmic reticulum (ER) and Golgi[2], an inhibitor targeting a specific disease-associated glycosyltransferase could be effective. However, structural information on glycosyltransferases is critically lacking, which has hampered attempts to design novel glycan-based drugs.

*N*-acetylglucosaminyltransferase-V (GnT-V), encoded by the *MGAT5* gene, is one of the most characterized cancer-associated glycosyltransferases[10,11]. GnT-V transfers an *N*-acetylglucosamine (GlcNAc) residue from high-energy donor UDP-GlcNAc to the α1-6-linked mannose of *N*-glycans via a β1-6 linkage[11,12] (Fig. 1a). Elevated expression of this β1-6 GlcNAc branch is observed in various cancers and correlates well with cancer malignancy and poor prognosis[13,14]. Consistent with this, mRNA levels of *MGAT5* gene are upregulated in various cancer types by direct transcriptional activation of the oncogenic Ras-

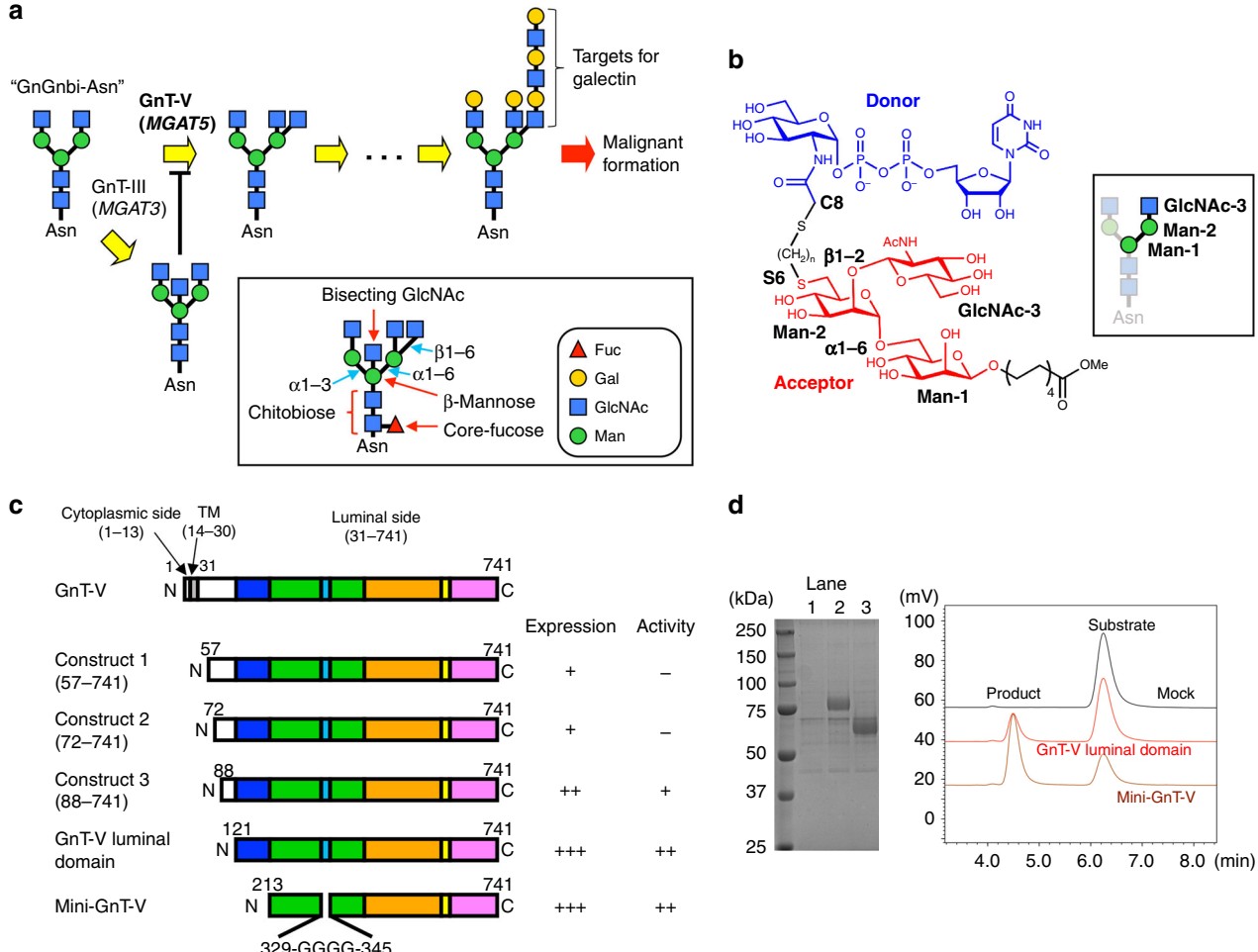

**Fig. 1** GnT-V transfers β1-6 branch on *N*-glycan via C-terminal region. **a** The β1-6 branch formation of *N*-glycan catalyzed by GnT-V. The substrate of GnT-V is GlcNAc-terminated biantennary glycan termed "GnGnbi-Asn". The β1-6 branch formation leads to further LacNAc extension, which is the target for galectin. The β1-4 branch at β-mannose catalyzed by GnT-III, named as bisecting GlcNAc, hampers β1-6 branch formation. Monosaccharide symbols follow the symbol nomenclature for glycans (SNFG) system[79]. Nomenclature of sugar residues (red arrows) and glycosidic linkages (cyan arrows) are indicated in boxes. **b** Chemical structure of bisubstrate-type inhibitor of GnT-V used for co-crystallization. Donor and acceptor substrates are colored with blue and red, respectively. Three sugar residues (GlcNAcβ1-2Manα1-6Man) of acceptor moiety are labeled. In this study, the inhibitor of which the linker length is ($n = 3$) and $K_i$ value is 18 μM[25] was used. Acceptor trisaccharide corresponds to α1-6 branch of biantennary glycan (GnGnbi) as shown in the box. **c** Domain architecture of full length and truncated constructs of human GnT-V. Expression levels and catalytic activities of truncated constructs are indicated on the right of these constructs. **d** Enzymatic activity of GnT-V luminal domain and mini-GnT-V. The two constructs were expressed in COS-7 cells and purified through Ni²⁺-column, followed by SDS-PAGE with CBB staining (left) or activity assays (right). Lane 1, mock, lane 2, GnT-V luminal domain, and lane 3, mini-GnT-V. The eluates from Ni²⁺-column were incubated with acceptor substrate GnGnbi-PA and donor substrate UDP-GlcNAc, and the reaction mixtures were analyzed by HPLC

Raf-Ets pathway[15,16]. Furthermore, genetic ablation of the *MGAT5* gene in mice resulted in reduced growth and metastasis of a mammary tumor[17], and overexpression of GnT-V in three-dimensional (3D)-cultured mammary normal epithelial cells caused neoplastic change with abnormal cancerous cell morphology[18]. Mechanistically, the β1-6 branch glycan modification by GnT-V allows target proteins to be decorated with a number of *N*-acetyllactosamine (LacNAc) residues, creating high-affinity ligands for galectins[19] (Fig. 1a). Glycan–galectin interaction prolongs the appearance of growth factor receptors at the cell surface, leading to enhanced and prolonged growth signaling[20]. Furthermore, formation of the β1-6 branch on cadherins and integrins impacts on their adhesive properties and inhibits cell adhesion and enhances cell migration[21–23], aiding cancer invasion and metastasis. Interestingly, GnT-V activity toward several substrate glycoproteins is *N*-glycosylation site specific[22,24]. Accumulating evidence strongly suggests that the β1-6 branch produced by GnT-V promotes both cancer initiation and progression, and that GnT-V inhibitors are feasible drug candidates. Although we previously developed a bisubstrate-type inhibitor of GnT-V, which contains both an acceptor trisaccharide unit (GlcNAcβ1-2Manα1-6Man) and a donor UDP-GlcNAc component (Fig. 1b) and shows moderate inhibition of GnT-V ($Ki = 18.3$ μM)[25], there is a need for more potent and specific inhibitors in clinical treatment.

Despite its functional importance, the 3D structure of GnT-V, as well as the mechanism of catalysis remain unsolved. GnT-V acts only on the α1-6 mannose branch of *N*-glycans (Fig. 1a), and this process is highly regulated. For instance, formation of another GlcNAc branch (bisecting GlcNAc) by GnT-III completely interferes with GnT-V action[11] (Fig. 1a). This regulated modification by GnT-V underscores the enzymatic basis of the complex and specific function of glycans, but as yet the underlying catalytic mechanisms have not been clarified. Detailed structural information on GnT-V would greatly improve our understanding of its strict recognition of acceptor substrates, as well as have a positive bearing on therapeutic applications. Of note, in contrast to the other mammalian GlcNAc transferases (GnTs) such as GnT-I, II, III IVa, and IVb, GnT-V lacks the conventional Asp-x-Asp (DXD) motif, which is a critical sugar-binding motif commonly found in many vertebrate glycosyltransferases, and its catalytic reaction is metal independent, with weak donor binding and tight acceptor binding[10] (Supplementary Figure 1). These differences highlight the uniqueness of GnT-V.

In this study, we determined the crystal structures of the GnT-V catalytic domain in an apo form and in complex with our bisubstrate-type inhibitor as a substrate analog. We pin-point catalytically important amino-acid residues by mutational experiments and propose a catalytic reaction mechanism for GnT-V.

## Results

### Expression and catalytic activity of truncated GnT-V. 
Human GnT-V is a type II membrane protein and comprises 741 amino acids in total. The luminal domain of GnT-V was originally predicted as H31-L741. We tested a series of constructs truncating the N-terminal region and found T121-L741, named GnT-V luminal domain, is sufficient for catalytic reaction without impairing protein production in mammalian cells (Figs. 1c, d). Furthermore, we designed a novel truncated enzyme, designated as mini-GnT-V, for structural determination of the complex with the bisubstrate-type inhibitor as described in a later section. Mini-GnT-V was also successfully expressed in mammalian cells and shown to retain full enzymatic activity (Fig. 1d).

### Overall structure of GnT-V luminal domain. 
We determined the crystal structure of GnT-V luminal domain (T121-L741) in apo form at 1.9 Å resolution by the native single-wavelength anomalous diffraction (SAD) method (Fig. 2a). Overall structure of GnT-V luminal domain has a cross-like shape and is composed of four subdomains: N-terminal domain (A128-Y207 in blue and G337-P339 in cyan in Fig. 2a), middle domains 1 (H212-K329 and I345-L424 in green, and E606-H627 in yellow) and 2 (G425-Y605 in orange), and C-terminal domain (G628-L741 in pink). As both of the two middle domains take Rossmann folds, GnT-V is classified into the GT-B fold. This is consistent with the previous prediction by homology modeling[26] and is in marked contrast to other mammalian GlcNAc transferases such as GnT-I (GT13)[27], GnT-II (GT16)[28], POMGnT1[29], and C2GnT-L[30], which are classified into GT-A folds (Supplementary Figure 1). The GT-A fold contains a single Rossmann-fold and usually has the DXD motif, which is critical for binding a divalent cation. As the other human GnTs also require a divalent cation for catalysis, GnT-V is unique among human GnTs. The four subdomains of GnT-V luminal domain do not exist independently but instead are tightly connected by two short insertions (shown in cyan and yellow in Fig. 2a), as well as interregional disulfide bonds such as C172-C338 and C372-C626 (Fig. 2b and Supplementary Figure 3). And, the N-terminal domain is linked to middle domain 1 via three missing loop regions, signifying the highly mobile nature of this domain (Fig. 2a). GnT-V luminal domain contains eight disulfide bonds in total and five of them are located at the C-terminal domain. GnT-V was reported to have three *N*-glycans at N334, N433, and N447[31]. In the crystal structure, only one *N*-glycan was assigned at N433 (Fig. 2a). The N334 is located in a disordered loop, whereas N447 is visible but the *N*-glycan is missing probably due to high mobility.

### Catalytic activity of GnT-V luminal domain. 
GnT-V belongs to the GT18 family in the CAZy database[32] and up until now no crystal structure of this family has been elucidated. A DALI search[33] shows GnT-V has structural similarities to bacterial glycosyltransferases in the GT4 family such as first mannosyltransferase Wbaz-1 (PDB code: 2F9F, $Z$-score = 12.4), *Ba*BshA, also referred to as *Ba*GT4$_{BA1558}$ (PDB code: 2JJM, $Z$-score = 12.0) and phosphatidylinositol mannosyltransferase (PDB code: 2GEJ, $Z$-score = 11.8). Although all these members are retaining enzymes, *Ba*BshA also uses UDP-GlcNAc as donor substrate, as in the case of inverting GnT-V (Supplementary Figure 4A). Structural superposition of *Ba*BshA (PDB code: 3MBO[34]) shows a root mean square deviation (RMSD) of corresponding 262 Cα atoms of 3.8 Å. The position of E526 in GnT-V coincides well with that of E290 (E291) in the *Ba*BshA-UDP complex (Supplementary Figure 4B). This glutamate of *Ba*BshA directly interacts with the ribose moiety of the donor substrate. E520 is also located close, but this residue is slightly buried and forms a tight salt bridge with R558. In addition, several residues of *Ba*BshA, which interact with UDP-GlcNAc such as N206, K211, and E282, are not conserved in GnT-V. The lack of polar residues may be a reason for the weak donor binding of GnT-V, compared with other human GnTs[35,36] (Supplementary Figure 1).

Inverting glycosyltransferases catalyze an $S_{N2}$ reaction in a single displacement step with the formation of an oxocarbenium ion-like transition state. The reaction is assisted by a catalytic base, usually an acidic residue such as aspartate or glutamate, which deprotonates the incoming electrophile of the acceptor[37]. In GT-B folds, the catalytic center often resides in a large cleft between two Rossmann-fold domains[38]. In addition, the catalytic residue of GnT-V is likely to be located around the

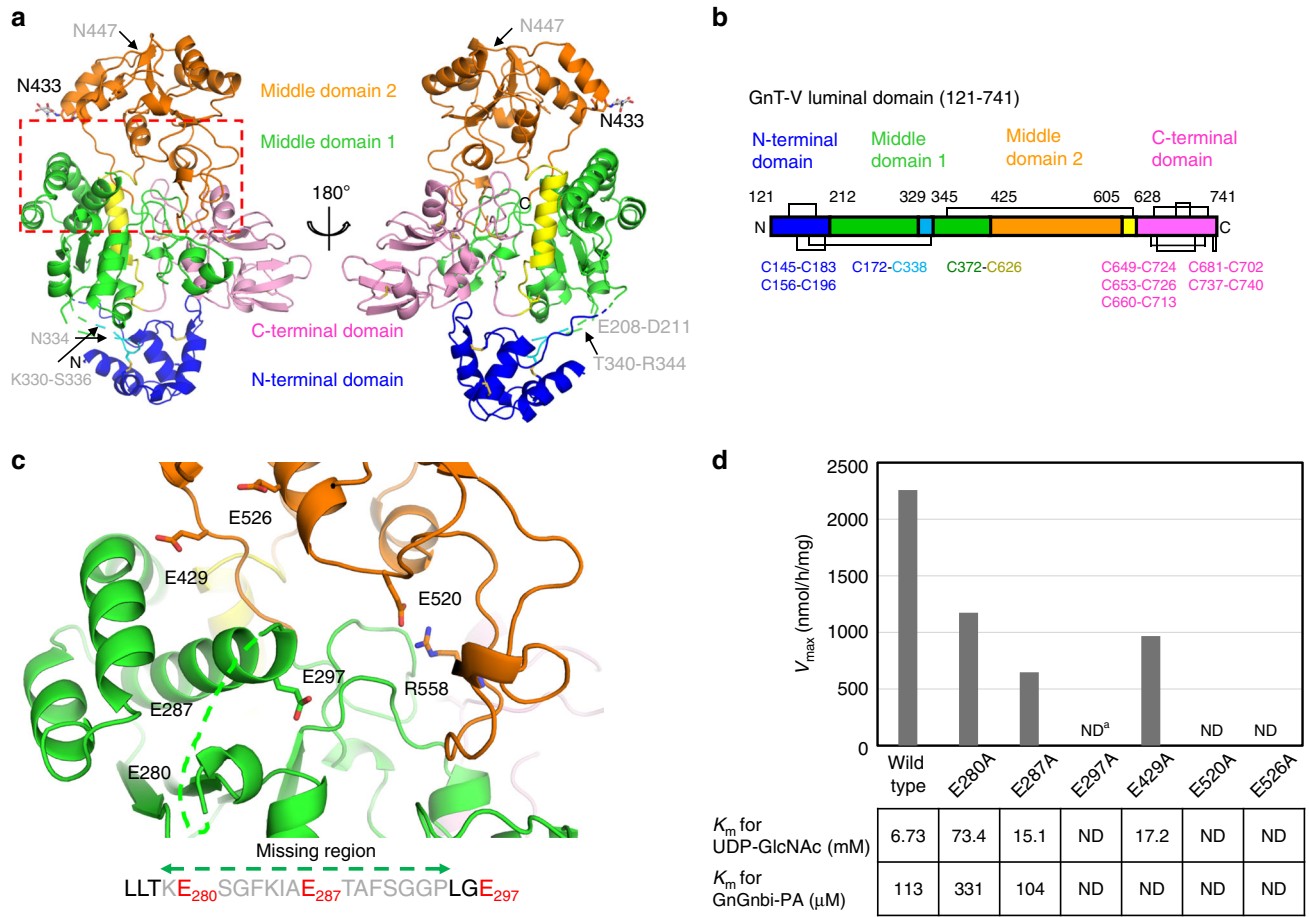

**Fig. 2** Overall structure and catalytic residues of GnT-V luminal domain. **a** Overall structure of GnT-V luminal domain is shown in ribbon model. The N-terminal domain (A128-Y207 in blue and G337-P339 in cyan), middle domain 1 (H212-K329 and I345-L424 in green and E606-H627 in yellow), middle domain 2 (G425-Y605 in orange), and C-terminal domain (G628-L741 in pink) are shown in ribbon models. The disulfide bonds and *N*-glycan attached on N433 are indicated with rod models and labeled. Putative catalytic center of GnT-V is indicated with a red dotted box. Among three putative *N*-glycans (N334, N433, and N447), only GlcNAc residue of N433 is assigned in the final model. Three *N*-glycosylation sites are indicated. The three missing loops between N-terminal domain and middle domain 1 are also indicated. **b** Schematic representation of disulfide bond pattern of GnT-V luminal domain. Cysteine residues, which form disulfide bonds, are also indicated. **c** Structural positions of six glutamates (E280, E287, E297, E429, E520, and E526) in GnT-V in respect of mutational experiments are shown in rod model. E280 and E287 are located in the missing loop region (K279-P294) shown as green dotted line. Sequence of missing loop region is shown in lower panel. The side chain of R558, which forms a salt bridge with E526 is also shown in rod model. **d** Kinetic parameters of wild-type and mutated GnT-V luminal domains. $V_{max}$ values are shown in bar graph representation. $K_m$ values for UDP-GlcNAc (donor substrate) and GnGnbi-PA (acceptor substrate) are shown in lower panel. [a]ND not detected

corresponding UDP- and GlcNAc-binding sites of *Ba*BshA (Supplementary Figure 4), suggesting that one of the six glutamates (E280, E287, E297, E429, E520, and E526 in Fig. 2c) of GnT-V between the two middle domains is a catalytic base. Among these residues, E280 and E287 are located in the missing loop (bottom line in Fig. 2c). We measured the catalytic activities of alanine mutants of these residues as well as wild-type enzyme (Fig. 2d). Mutation of three glutamates (E297, E520, and E526) individually severely impaired enzymatic activity. Although the three mutants lost enzymatic activity equally in the usual assay with a 1-h reaction time, prolonged incubation (10 h) of a greater amount of E520A mutant showed slight but measurable activity, whereas E297A and E526A did not (Supplementary Figure 5). These results show that two glutamates, E297 and E526, are indispensable for catalysis. E297 is located at the lid of the cleft formed by the two middle domains and is likely well positioned to act as a catalytic base. As suggested from structural comparison with *Ba*BshA, E526 of GnT-V may also bind donor substrate and the alanine mutant likely impairs binding.

**Overall structure of mini-GnT-V in complex with inhibitor.** To obtain complex structures with donor and acceptor substrates, we tried to crystallize wild-type, E297A, E520A, or E526A GnT-V luminal domain in complex with donor UDP-GlcNAc or UDP and/or acceptor *N*-glycans by co-crystallization, as well as soaking methods. Among the three mutants, E297A was the only one, which allowed successful crystallization even in the presence of ligands, although no clear electron density of substrates was observed. As a result of these trials, we decided to change the construct. N-terminal domain (A128-Y207) is independent of the catalytic GT-B fold and seems unnecessary for catalysis (Fig. 2a). Consequently, we designed a new truncated construct of GnT-V luminal domain, mini-GnT-V, in which N-terminal domain (T121-N212) was deleted. Also, the long unstructured loop (K330-R344), involved in an interregional disulfide bond (C172-C338), was replaced with a short linker of four glycine residues to reduce the structural heterogeneity (Fig. 1c). The expression level and enzymatic activity of mini-GnT-V were comparable to those of the GnT-V luminal domain (Fig. 1d). We further introduced the inactive E297A mutation to mini-GnT-V to form a stable

complex with ligand. We used our own bisubstrate-type inhibitor as ligand. Crystal structure of mini-GnT-V E297A mutant in complex with the ligand was successfully determined at 2.1 Å resolution (Fig. 3a). The asymmetric unit contains two mini-GnT-V inhibitor complexes, referred to as complex A and B. Structural superposition of the two complexes demonstrated that overall RMSD value of Cα atoms is only 0.76 Å. Of note, one difference is that the weak electron density of a highly mobile region (G282-G296) is observed in complex B and included in the final model (Fig. 3a). This loop is missing in complex A, as well as in the apo form. Structural differences between the apo form and the inhibitor complex are also small. Compared with the apo form structure, RMSD values of Cα atoms of complexes A and B are 0.98 and 1.16 Å, respectively (Supplementary Figure 6A). The deletion of N-terminal domain and long loop (K330-R344) do not affect the local structure around middle domain 1 as intended (Fig. 3b).

Although the electron densities of donor and linker moieties were obscure, the electron density of acceptor trisaccharide moiety, GlcNAcβ1-2Manα1-6Man, was clearly visible (Fig. 3c). The trisaccharide moiety binds to mini-GnT-V between the two

middle domains (Fig. 3d). The interaction modes of the two complexes are almost the same (Supplementary Table 2). Among the three sugar residues, both GlcNAc-3 and Man-2 are buried in the cavity and tightly bind to mini-GnT-V via direct and water-mediated interactions, whereas Man-1 is exposed to solvent (Fig. 3d and Supplementary Table 2). The OH3 of GlcNAc-3 binds to the backbone and side chain of S379, whereas OH4 interacts with the side chain of D378. The side chain of K554 interacts with O5 and S6 (OH6) of GlcNAc-3, as well as OH3 of Man-2. Three water molecules link mini-GnT-V to GlcNAc-3 by interacting with OH3, OH4, and OH6. Notably, two aromatic rings of F380 and W401 contact the trisaccharide and the side chain of W401 and seem to restrain the conformation of the α1-6 branch (Fig. 3e). The dihedral angles of the Manα1-6Man linkage in the trisaccharide are categorized as extend-a conformation, which is the preferred conformation of biantennary glycan[39] (Table 1). In addition, the dihedral angles of GlcNAcβ1-2Man in the acceptor complex coincide well with previous solution NMR analysis[40] (Table 1), indicating the structure of this disaccharide unit is firmly stabilized by GnT-V. Compared with the apo form structure, the side chains of F380 and W401 are flipped and seem

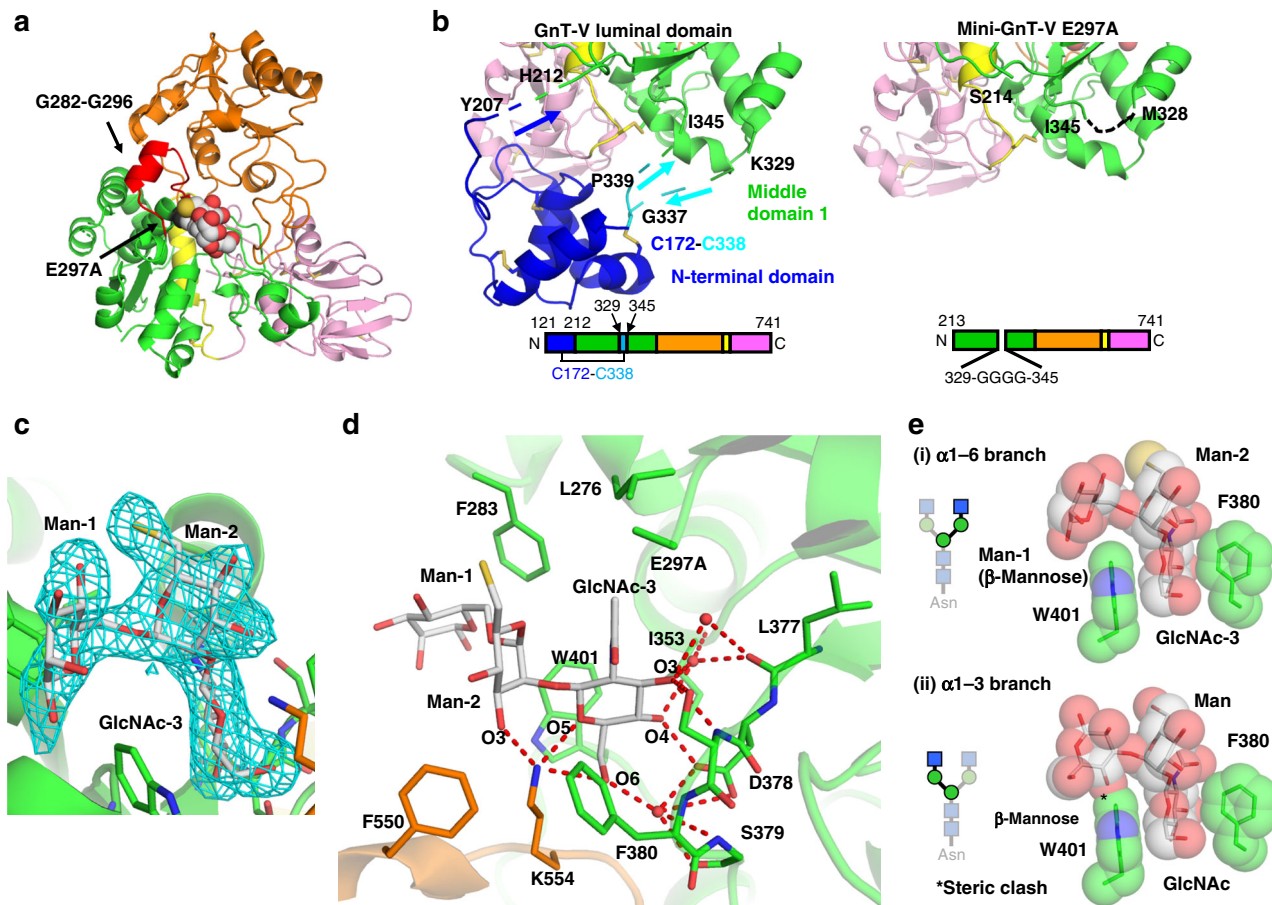

**Fig. 3** Structure of mini-GnT-V E297A in complex with bisubstrate-type inhibitor. **a** Crystal structure of mini-GnT-V E297A in complex with bisubstrate-type inhibitor. Trisaccharide is shown in sphere model. Highly mobile region (G282-G296) and E297A are colored with red and black, respectively, and indicated. **b** Close-up views of domain boundary between N-terminal domain and middle domain 1 in GnT-V luminal domain (left panel) and mini-GnT-V E297A (right panel). Key residues of these two structures are labeled. Domain architectures are shown in bottom. **c** Omit map contoured at 3.0 σ level around acceptor binding site is shown in cyan mesh. Carbohydrate and amino-acid residues are shown in rod models. **d** Close-up view of interaction with acceptor trisaccharide. Direct and water-mediated interaction network is depicted with red dotted lines. Three water molecules, which link glycan and mini-GnT-V are conserved in the two complexes. Details of interaction network are also summarized in Supplementary Table 2. **e** Two aromatic residues, F380 and W401, restrict the conformation of trisaccharide and define the branch specificity. Two aromatic rings and trisaccharide residues are shown in rod and semi-transparent sphere models. The interaction of α1-6 branch observed in crystal structure (i) and docking model of α1-3 branch (ii) are shown in upper and lower panels, respectively. Steric clash is shown in asterisk

**Table 1 Dihedral angles of acceptor glycan compared with previous NMR analysis[40]**

| Linkage | Complex A | Complex B | trNOE analysis[40] |
|---|---|---|---|
| GlcNAcβ1-2Man ($\phi_{\beta1-2}$, $\varphi_{\beta1-2}$) | (−84°, −80°) | (−82°, −89°) | (−107° ± 7, −84° ± 2) |
| Manα1-6Man ($\phi_{\alpha1-6}$, $\varphi_{\alpha1-6}$, $\omega_{\alpha1-6}$) | (59°, 165°, 64°) | (69°, 165°, 49°) | - |
| Manα1-6Glc ($\phi_{\alpha1-6}$, $\varphi_{\alpha1-6}$, $\omega_{\alpha1-6}$) | - | - | (116° ± 3, 119° ± 6, 60° ± 3) |
| Conformation | Extend-a | | |

The $\phi$, $\varphi$, and $\omega$ angles are defined by atoms O5–C1–O'x–C'x, C1–O'x–C'x–C'x-1, and O'x–C'x–C'x-1–C'x-2, respectively

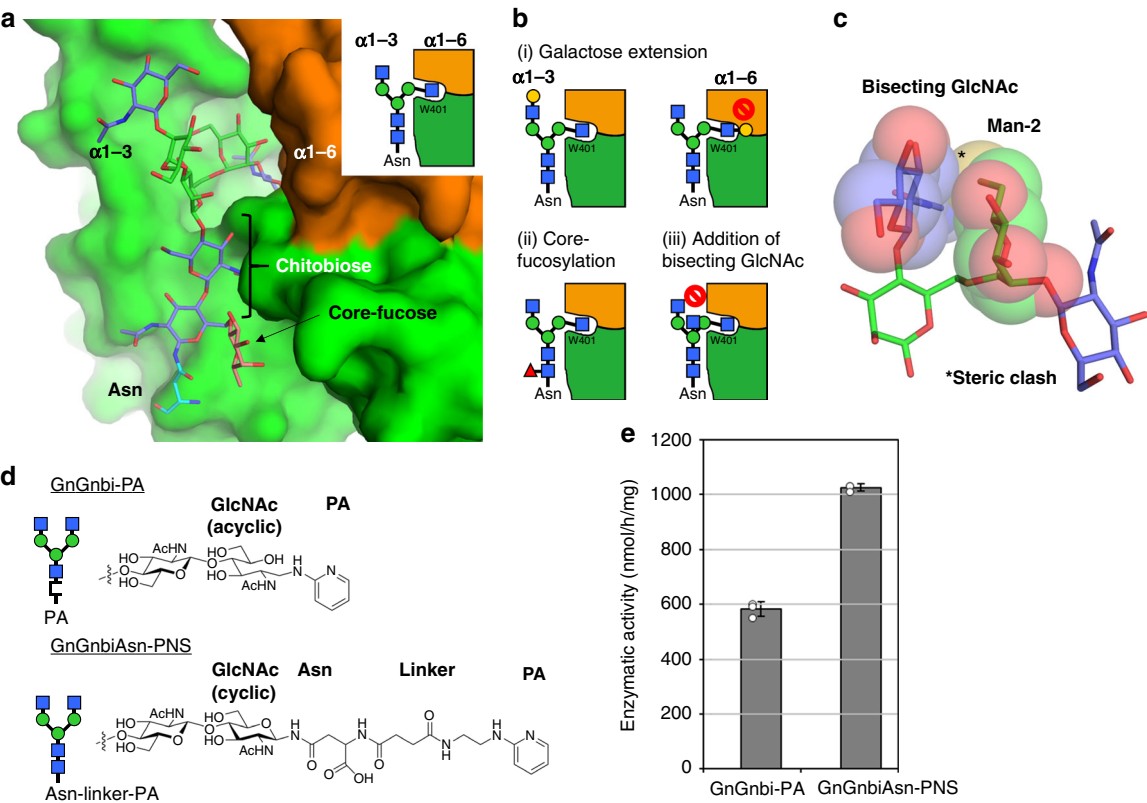

**Fig. 4** Substrate specificity of GnT-V toward various types of N-glycans. **a** Docking model of N-glycan bearing α1-3 branch, chitobiose, core fucose, and asparagine. Sugar residues and GnT-V are shown in rod and surface models, respectively. Schematic drawing of interaction mode is also shown in right panel. Structure of N-glycan was built based on the atomic structures of biantennary glycans (PDB codes: 5XFI[77] and 4BM7[78]). **b** Schematic drawing of interaction modes in various types of N-glycans. (i) Galactose extension at two branches, (ii) core fucosylation, and (iii) addition of bisecting GlcNAc are shown. Structural details of these interactions are also shown in Supplementary Figures 7 and 8. **c** Superposition of bisecting GlcNAc onto trisaccharide structure shown in rod model. Bisecting GlcNAc and Man-2 are also shown in semi-transparent spheres. Steric conflict is indicated with asterisk. **d** Two differently labeled biantennary N-glycans applied to enzymatic assays. The oligosaccharide-type glycan (GnGnbi-PA) and asparagine-type glycan (GnGnbi-Asn-PNS) are shown in upper and lower panels, respectively. Schematic drawing of labeled glycans and close-up view of chemical structures around chitobiose unit of two substrates are also shown in left panel. **e** Enzymatic activity of GnT-V luminal domain toward the two types of substrates (100 pmol each per reaction) is shown. Activity was repeatedly measured (n = 3) using the same enzymes and substrates (n = 3). The graph shows means ± S.D

to sandwich GlcNAc-3 upon acceptor binding (Supplementary Figure 6B).

A preferred sugar substrate of GnT-V in vitro is a GlcNAc-terminated biantennary glycan, GnGnbi (Fig. 1a). Although the disaccharide unit (GlcNAcβ1-2Man) is commonly found in both α1-3 and α1-6 branches, GnT-V transfers β1-6-linked GlcNAc to only the α1-6 branch and not to the α1-3 branch (Fig. 1a). Superposition of the trisaccharide unit of the α1-3 branch (GlcNAcβ1-2Manα1-3Man) shows that the β-mannose sterically clashes with the aromatic ring of W401 (Fig. 3e). This strongly suggests that W401 specifically selects the α1-6 branch. Besides, even when we chose a rotamer, which can fit into the active site, the chitobiose moiety sterically clashes with GnT-V (Supplementary Figure 7A).

The acceptor trisaccharide is deeply buried inside the cavity of GnT-V, raising the possibility that the surface groove around the cavity plays a role in the fine specificity toward various types of N-glycans. To investigate this, we built a docking model with the N-glycan bearing α1-3 branch, chitobiose and core-fucose (Fig. 4a). These sugar residues fit well into the groove on the protein surface. The model gives a plausible explanation for the strict substrate specificity of GnT-V indicated by previous biochemical studies. Galactose extension of either the α1-3 or α1-6 branch hampers GnT-V activity in enzyme assays[11]. As

GlcNAc-3 on the α1-6 branch is deeply buried in the binding site, extension of galactose at this branch would cause a serious clash with D378 (Fig. 4b and Supplementary Figure 7B), whereas galactosylation of the α1-3 branch creates no steric clash with the protein (Fig. 4b and Supplementary Figure 7B). Core fucosylation catalyzed by Fut8 does not impair β1-6 branch formation of GnT-V[11]. Consistent with this, a docking model shows that core fucosylation seems to have no effect on GnT-V binding (Figs. 4a, b). Bisecting GlcNAc synthesized by GnT-III completely inhibits GnT-V activity[23] (Fig. 1a). When a bisecting GlcNAc residue was put on β-mannose (Man-1) of trisaccharide, this residue lies close to OH6 of the α1-6 branched mannose (Man-2) and sterically clashes with Man-2 (Fig. 4c). In addition, bisected glycan prefers to take extend-b and back-fold conformations[39,41], rather than extend-a conformation. Structural superposition of these two conformations shows that they cannot fit into the surface groove of GnT-V without severe steric clashes (Supplementary Figure 8). These results explain why the introduction of bisecting GlcNAc prevents β1-6 branch formation.

The docking model suggests that chitobiose and the asparagine to which N-glycan is attached fit well into the narrow groove (Fig. 4a). To evaluate the interaction between the core region of N-glycan and GnT-V, we performed enzymatic assays using two different fluorescence-labeled substrates (Fig. 4d). Pyridylaminated (PA) N-glycan (GnGnbi-PA) has acyclic GlcNAc at the reducing end and no asparagine, whereas asparagine-type substrate (GnGnbi-Asn-PNS) contains both cyclic GlcNAc and an asparagine residue. Although the two substrates have the same N-glycan structure, the asparagine-containing substrate is the better one (Fig. 4e). This suggests that the core region of N-glycan does interact with GnT-V and the contacts contribute to substrate specificity,

although it is also possible that the attached linker chain may modify the interaction.

**Possible catalytic reaction mechanism of GnT-V.** Crystallographic and biochemical studies provide a rational catalytic reaction mechanism of GnT-V. Structural superposition of the unliganded form of wild-type GnT-V and the ligand complex of mini-GnT-V E297A shows that E297 is the only acidic residue within 5 Å of OH6 (S6) of Man-2 (Fig. 5a). This strongly suggests that E297 functions as a catalytic base, consistent with the mutational analysis (Fig. 2d). In comparison with the BaBshA structures, the UDP (or UMP) moiety of BaBshA ligand complexes fit into the groove of GnT-V without apparent clash, but the product GlcNAc-malate clashes with the loop (P518-P522) of GnT-V (Fig. 5b). Therefore, we built in donor substrate, UDP-GlcNAc, to the GnT-V-ligand structure, based on the superposition of UDP-GlcNAc extracted from GnT-I donor complex (PDB code: 1FOA[27]) onto UDP moiety position by least square fitting (Fig. 5c). In this GnT-V ternary complex model, the likely catalytic base E297 is located within hydrogen bond distance of OH6 of Man-2 (3.2 Å), but the distance between OH6 of Man-2 and C1 atom of UDP-GlcNAc is 6.0 Å, requiring a positional shift for efficient catalysis. In other GlcNAc transferases with GT-A folds, the catalytic base residues are commonly located close to the hydroxyl group of acceptor substrate (Supplementary Figure 9). In GnT-V-ligand structure, E297 is located at the base of a highly mobile loop (G282-G296), and it is conceivable that this loop, and the putative catalytic residue, moves to effect catalysis (Fig. 5a). Apart from E297, there are five other glutamates in the binding cleft, and all six were tested in mutational experiments (Fig. 2d). Glutamates, E429, E520, and E526 are close to the putative donor binding site; E526 is near the ribose moiety, and

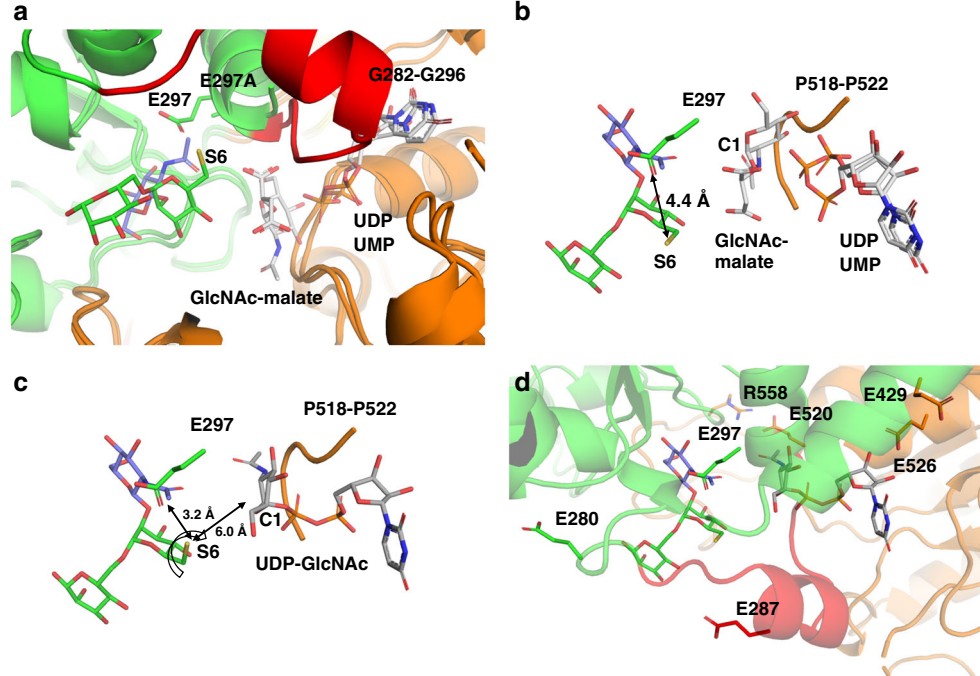

**Fig. 5** Putative ternary complex structure of wild-type GnT-V and substrates. **a** Structural superposition of GnT-V luminal domain (apo form), mini-GnT-V in complex with acceptor, BaBshA-UDP complex (PDB code: 3MBO) and BaBshA-UMP-GlcNAc-malate complex (PDB code: 5D00). **b** Close-up view of ligand binding site. The acceptor trisaccharide (mini-GnT-V-acceptor complex), UDP, UMP, and GlcNAc-malate (BaBshA ligand complexes) are shown. The distance between E297 and S6 of Man-2 is also indicated. **c** Ternary complex model of wild-type GnT-V, acceptor trisaccharide and UDP-GlcNAc. The position of UDP-GlcNAc was inferred by superposing the structure of UDP-GlcNAc extracted from GnT-I-UDP-GlcNAc complex (PDB code: 1FOA[27]) onto UDP moiety of BaBshA complex. The S6 of Man-2 is also rotated toward E297. The distances between S6 of Man-2 and E297 or C1 of UDP-GlcNAc are indicated. **d** Six glutamates selected for mutational experiments (Fig. 2c) are shown in the ternary complex

so is E429, though slightly further away in the present model, consistent with mutant E429A retaining enzymatic activity, whereas E520 lies close to GlcNAc and E520A severely impairs activity. This latter residue is slightly buried and forms a tight salt bridge with R558 and may also play a role in donor interaction. The other two glutamates, E280 and E287, are more apart from donor and acceptor substrates, and mutants E280A and E287A retain enzymatic activity, indicating only supportive roles (Figs. 2d and 5d).

**Possible mechanism of site-specific modification of GnT-V.** The β1-6 branch formation catalyzed by GnT-V regulates the physiological activities of substrate glycoproteins such as T-cell receptor, integrins, growth factor receptors, and cadherins[42,43]. The β1-6 branch formation of some substrates such as E-cadherin and CEACAM6 is known to occur in a glycosylation site-specific manner[22,24]. The site-specific modification of E-cadherin affects cellular localization and cell–cell contacts[22]. E-cadherin is a type I membrane protein and has five ectodomains (EC1–5) (Fig. 6a). The β1-6 branch formation of N-glycan attached onto N554 at EC4 was reported to critically regulate oncogenesis and malignancy of gastric cancer[22], whereas the N-glycan attached onto N633 at EC5 was not modified by GnT-V, showing that GnT-V somehow selects specific target N-glycans. The N-glycan at N633 is instead essential for structural stability rather than cell–cell communication, as the N633Q mutation impairs the protein stability[44]. N554 is positioned in the middle of a β-strand and the side chain is fully exposed to solvent, whereas N633 resides in the loop region and the side chain is slightly buried (Fig. 6b). Docking models with murine E-cadherin ectodomain (PDB code: 3Q2V[45]) demonstrate that GnT-V can access the N-glycan at N554 without severe steric clash (Fig. 6c), whereas GnT-V sterically clashes with the adjacent region of N633 (Fig. 6d). In addition, N633 is thought to reside close to the membrane (Fig. 6a), and it seems difficult for GnT-V to enter the narrow space between EC5 and

the membrane. Structural accessibility of the N-glycan on substrate glycoproteins to GnT-V may determine their potential for modification.

## Discussion

In this study, we clarified the unique structural features and atomic details of N-glycan recognition of GnT-V. GnT-V luminal domain takes a GT-B fold with two accessary domains. Compared with other GT18 family members, such as GnT-IX (also known as GnT-Vb), which is a paralog of GnT-V and C. elegans ortholog, gly-2, the cysteine residues for disulfide bonds are completely conserved, indicating these enzymes share the same structural features (Supplementary Figure 3). GnT-IX attaches β1-6-linked GlcNAc residue to O-mannosylated core M1 glycan (GlcNAcβ1-2Man-O-Ser/Thr), as well as N-glycan[46,47]. GnT-V and -IX share amino-acid residues involved in acceptor substrate binding such as D378, F380, W401, and K554, indicating the recognition mode of the disaccharide part, GlcNAcβ1-2Man, is common. Modeling analysis shows core M1 glycan can also fit into the acceptor binding site of GnT-V (Supplementary Figure 10A). This is consistent with a previous analysis showing that GnT-V partially compensates for GnT-IX activity in vivo[48]. Unlike GnT-V, GnT-IX prefers O-mannosylated glycan rather than N-glycan and is active even in the presence of a high concentration of manganese ions. In addition, GnT-IX is strongly inhibited by the bisubstrate-type inhibitor with a shorter linker[25] (n = 1 in Fig. 1b). Structure-based sequence comparison demonstrates that seven amino-acid residues of GnT-V, F283, K284, I353, V354, P400, and I552, around the acceptor binding site are replaced with different types of amino-acid residues in GnT-IX (Supplementary Figure 10B). All or some of these residues likely contribute to the different substrate specificities of the two enzymes.

N-terminal domain (A128-Y207 and G337-P339) is independent of the catalytic GT-B fold and we clearly demonstrated that

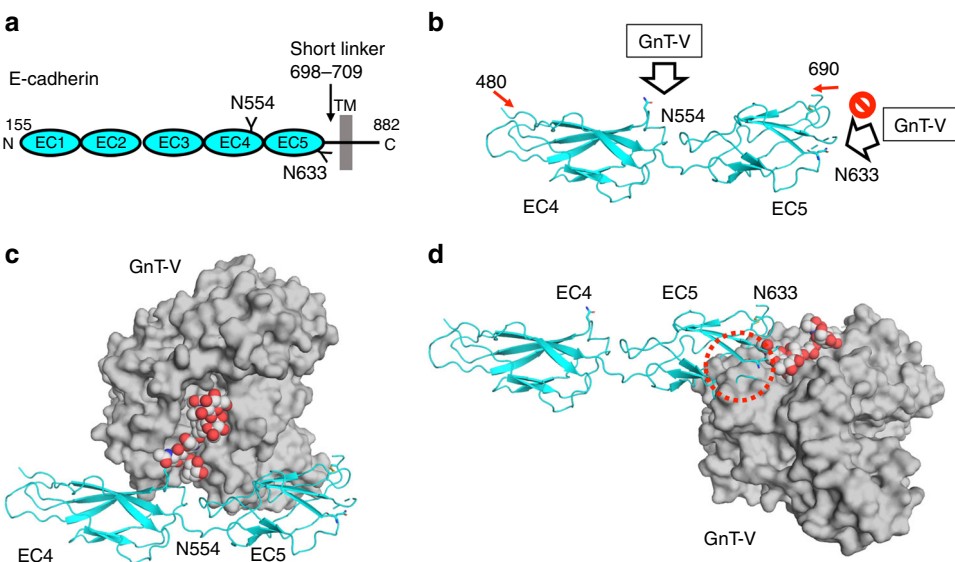

**Fig. 6** Hypothetical interaction mode between GnT-V and E-cadherin. **a** Schematic representation of human E-cadherin. E-cadherin is a type I membrane protein and has five ectodomains (EC1–5). Two N-glycans at N554 and N633 are highlighted. EC5 connects to transmembrane helix via short linker region (698-709). **b** Crystal structure of murine E-cadherin EC4-EC5 ectodomains (PDB code: 3Q2V[45]) is shown in ribbon model. Two N-glycosylation sites, N554 and N633, are shown in rod models. **c** Docking models of GnT-V and N-glycan at N554 of E-cadherin EC4-EC5 ectodomain. GnT-V, N-glycan and E-cadherin are shown in surface, sphere and ribbon models, respectively. **d** Docking models of GnT-V and N-glycan at N633 of E-cadherin EC4-EC5 ectodomains. GnT-V, N-glycan and E-cadherin are shown in surface, sphere and ribbon models, respectively. Steric clash is indicated with red dotted circle. The three figures, **b**–**d**, are depicted from the same view angle

mini-GnT-V although lacking the N-terminal domain retains full enzymatic activity and is properly folded (Figs. 1d and 3a). A CHO cell line having the L188R mutation in GnT-V is known as the Lec4A mutant, which decreases the surface expression of β1-6 branched N-glycan[49,50]. The corresponding L189 in human GnT-V is located in the hydrophobic core of the N-terminal domain and apart from the catalytic domain (Supplementary Figure 11). The impaired formation of β1-6 branch in the GnT-V Lec4A mutant was reported to be caused by mislocalization of GnT-V in the ER but not in the Golgi apparatus, rather than inactivation of catalytic activity[49,50]. The replacement of leucine with arginine, therefore, may result in improper folding of the N-terminal domain, leading to entrapment in the ER by the quality control system. Alternatively, the N-terminal domain may be actively involved in protein transport. As the short cytoplasmic tail of a glycosyltransferase is generally considered to determine its Golgi localization[51–53], GnT-V is likely equipped with a unique trafficking system regulated by the N-terminal domain on the luminal side. A DALI search demonstrated that the N-terminal domain has no apparent structural similarities with known mammalian proteins, but instead shows weak structural similarities with viral coat proteins such as SIRV (PDB code: 3F2E[54]) and AFV1 (PDB code: 3FBL[55]). Elucidation of the physiological function of this accessary domain of GnT-V will be an interesting issue to be solved in the future.

We applied the bisubstrate inhibitor for co-crystallization, but the electron density of the donor moiety was completely missing (Fig. 3c). Bisubstrate analogs, in which donor and acceptor analogues are covalently attached to each other, offer promise as both potent and selective inhibitors of glycosyltransferases[56]. As for GnT-V and IX, a series of bisubstrate-type inhibitors, which differ in linker length between S6 of Man-2 and C8 of GlcNAc were developed[25] ($n = 1$–3 in Fig. 1b). Among them, the inhibitor with the longest linker ($n = 3$) shows the most potent inhibitory activity against GnT-V. Even so, it is possible that the donor moiety does not properly fit the cavity due to the linker length still not being long enough. Or else, the donor moiety may be degraded during crystallization since it took 2 months to obtain diffraction quality crystals. Note though that a weak donor interaction is characteristic of GnT-V (Supplementary Figure 1), and the substitution with donor derivatives is another promising approach to tight binding inhibitors.

The structure complexed with acceptor well explains the specificity toward various types of N-glycans, though GnT-V interacts with only two sugar residues. Several modifications such as the addition of galactose and bisecting GlcNAc strongly impair enzymatic activity. We speculate that the high specificity is derived principally from the narrow and deep binding cavity of GnT-V and steric hindrance of the entry of unpreferable substrates. Limited access to the catalytic center may also affect the site-specific modification of N-glycans on E-cadherin and other glycoproteins (Figs. 6c, d). It may underlie regulation of selective glycosylation by GnT-V as in polysialylation in St8SiaIV[57]. Site-specific glycoproteomics confirms that the accessibility of asparagine correlates with N-glycan branch formation[58], and the docking model of GnT-V-E-cadherin complex at N633 supports this (Fig. 6d). Further docking simulation using the 3D structures of glycosyltransferase complexes may, in the future, allow predictions of glycan structures of a target protein.

Although the luminal domain of GnT-V was originally assigned to H31-L741, we excluded H31-T120 in this structural analysis. The residues M45-I187 of GnT-V, known as the stem region, are responsible for oligomeric association and Golgi localization[59]. Gel filtration analysis indicates that the GnT-V luminal domain and mini-GnT-V likely exist as a monomer in solution (Supplementary Table 3). Glycosyltransferases often form homo-oligomers[60]. GnT-V may oligomerize via the stem region and not the catalytic domain (Supplementary Figure 12A). As GnT-V transfers β1-6-linked GlcNAc to multidomain glycoproteins like cadherins and cell surface receptors such as integrins and growth factor receptors, a long stem region may facilitate access these various target glycoproteins.

Part of the GnT-V population is secreted into the extracellular space by cleavage at the stem region[61]. Secreted GnT-V may function not as an enzyme but rather as an angiogenic factor[61,62], and the local positively charged loop (K254-K269) in the middle of domain 1 has been suggested to be involved in the interaction with negatively charged proteoglycan[62] (Supplementary Figure 12B). This region is apart from the catalytic center and forms a tunnel, which has enough space for interacting with polysaccharide (Supplementary Figure 12C).

In summary, we have elucidated atomic details of the catalytic domain of cancer-associated human glycosyltransferase, GnT-V. The acceptor N-glycan specificity of GnT-V is now understood and thus the structural basis of N-glycan processing in eukaryotic cells. Furthermore, the docking model suggests that spatial accessibility of N-glycans of substrate glycoproteins to the GnT-V active site likely defines the potential for site-specific modification. Recently a glycomimetic compound was reported to specifically inhibit GnT-V activity and suppress glioblastoma invasion[63]. Structural details of the catalytic center of GnT-V provide direct information for improvement in design of such inhibitors. Collectively, our present findings will hopefully facilitate the development of potent and specific glycan-targeted drugs and provide us with a deeper understanding of the complex and elaborate glycosylation machinery in mammals.

## Methods

**Materials**. The bisubstrate-type inhibitor, which contains an acceptor trisaccharide (GlcNAcβ1-2Manα1-6Man) and a donor UDP-GlcNAc connected by a short linker, was chemically synthesized[25] and used as ligand in this study (Fig. 1b). The asparagine-type substrate, GnGnbi-Asn-PNS ([N-(2-(2-pyridylamino)ethyl)succinamidyl]), was prepared by labeling egg yolk-derived GnGnbi-Asn with PNSNB [N-(2-(2-pyridylamino)ethyl)succinamic acid 5-norbor-nene-2,3-dicarboxyimide ester][64]. The labeled substrate was purified by high-performance liquid chromatography (HPLC) with a reversed-phase and amide column.

**Plasmid DNA construction**. Nucleotide sequences of all primers used in this study are summarized in Supplementary Table 1. For large-scale cultivation, DNA fragments encoding human GnT-V luminal domain (T121-L741), and mini-GnT-V (N213-K329-G₄-I345-L741) E297A mutant were amplified by PCR and incorporated into pSec-nPA vectors as fusion protein with PA tag and tobacco etch virus (TEV) protease site[65]. DNA encoding mini-GnT-V E297A mutant was modified from the GnT-V luminal domain in three steps: first, the N-terminal region (T121-H212) of GnT-V luminal domain was removed by PCR using primers (N_del_fwd and rev in Supplementary Table 1), phosphorylated by T4 polynucleotide kinase (New England BioLabs) and ligated by Ligation high version 2 (Toyobo). Second, the loop region (K330-R344) was exchanged to four consecutive glycine residues by PCR using primers, loop change fwd and rev in Supplementary Table 1. Finally, E297A mutant was introduced by using QuickChange Lightning Site-Directed Mutagenesis Kit (Agilent) according to the manufacturer's protocol. For enzymatic assays, DNA fragments were incorporated into pcDNA-IH vectors as fusion proteins with N-terminal hexahistidine tags[66].

**Purification of GnT-V luminal domain and mini-GnT-V**. PA tag fused GnT-V luminal domain and mini-GnT-V E297A mutant were transiently expressed with the Expi293 expression system (Thermo Fisher Scientific) in the presence of 5 μM kifunensine. Cultivated supernatant was collected after 90 h and applied to NZ-1 antibody immobilized column. Protein-trapped NZ-1 column was washed with Tris-buffered saline (TBS) and the target protein was eluted with PA peptide. After removal of PA tag by TEV protease cleavage and deglycosylation by Endo H (New England BioLabs), the reactant was applied to a size exclusion chromatography using Superdex200 column (GE Healthcare). Eluted protein was concentrated by Amicon Ultra (molecular weight cut off: 10 kDa) up to 7–10 mg/mL.

**Crystallization of GnT-V luminal domain and mini-GnT-V**. All crystallization trials were performed by the sitting-drop vapor diffusion method at 293 K. Initial

crystallization conditions were searched using screening kits of Classic II, Protein Complex Suite and PEGs suite II (Qiagen).

Diffraction quality crystals of GnT-V luminal domain were obtained under the condition of 0.15 M DL-malic acid (pH 7.0) and 20 % (w/v) polyethylene glycol (PEG) 3350 (Classic II No. 91). Prior to data collection, crystals were soaked in reservoir medium containing 20 % (v/v) PEG 400 and then rapidly frozen by liquid nitrogen. For complex formation, purified mini-GnT-V was mixed with bisubstrate-type inhibitor solution at a final concentration of 1 mM. Protein concentration of mini-GnT-V was adjusted to 5 mg/mL. Crystals of mini-GnT-V in complex with bisubstrate-type inhibitor were obtained under the condition of 0.1 M Hepes (pH 7.0), 0.1 M magnesium chloride, and 15 % (w/v) PEG 4000 (PEGs II No.24). Prior to data collection, crystals were directly frozen by liquid nitrogen.

Diffraction experiments were performed at BL44XU (for GnT-V luminal domain apo form) in SPring-8 (Harima, Japan), BL-1A (for GnT-V luminal domain Sulfur-SAD), and AR-NW12A (for mini-GnT-V E297A inhibitor complex) in Photon Factory (Tsukuba, Japan). The data collection of all datasets was performed at the cryogenic temperature (100 K). The intensities of diffraction spots were integrated and scaled by XDS[67] and AIMLESS[68]. Initial phase determination of GnT-V luminal domain was performed by the single anomalous dispersion method using intrinsic sulfur atoms. The diffraction data covering 1080° range in total was obtained from three different points of single crystal. The positions of 20 out of 35 sulfur atoms were assigned during phase determination step. Initial model included 450 out of 621 amino-acid residues. The R/R$_{free}$ values of initial model were 39/41%. The phase determination and initial model building of GnT-V luminal domain were conducted by phenix.autosol of Phenix program suite[69] with throughout option. The phase determination of mini-GnT-V inhibitor complex was performed by molecular replacement using MOLREP[70]. Further model building was conducted manually using Coot[71]. Refinement was initially performed with Refmac5[72] and phenix.refine of Phenix program suite for final models. The models were validated with Molprobity[73]. The percentages of amino-acid residues in favored, allowed, and outlier regions in Ramachandran plots were as follows: favored/allowed/outlier = 98.2/1.8/0% (GnT-V luminal domain apo form) and 97.1/2.9/0% (mini-GnT-V E297A inhibitor complex). Data collection and refinement statistics are summarized in Table 2. The quality of electron density map of refined structures is shown in Supplementary Figure 2. Atomic coordinates and structure factors of GnT-V luminal domain and mini-GnT-V inhibitor complex were deposited in Protein Data Bank under accession codes 5ZIB and

5ZIC, respectively. All figures were prepared with PyMOL (Schrödinger). Structural superposition was performed with SUPERPOSE[74] and LSQKAB[75]. The distances between GnT-V and sugar residues were investigated using NCONT of CCP4 program suite[76].

**Construction of docking model.** The docking model of GnT-V in complex with biantennary glycan bearing α1-3 branch and chitobiose was built as follows: The position of β-Mannose (Man-1) in mini-GnT-V-trisaccharide complex was superposed onto the corresponding residue of biantennary glycan (Galβ1-4GlcNAcβ1-2Manα1-3(GlcNAcβ1-2Manα1-6)Manβ1-4GlcNAcβ1-4GlcNAc, PDB code: 5XFI[77]) by LSQKAB[75]. The positions of asparagine and core fucose were further docked onto the crystal structure of N-glycan of IgG Fc (PDB code: 4BM7[78]) based on the position of the reducing end of chitobiose. The docking models of GnT-V and N-glycosylated E-cadherin complexes were built by super-posing GnT-V-N-glycan complex onto N404 and N483 of crystal structure of murine E-cadherin (PDB code: 3Q2V[45]). The N404 and N483 in PDB file correspond to N554 and N633 in human E-cadherin (Genbank: L08599), respectively.

**Enzymatic assay of wild-type and mutated GnT-V.** N-terminal histidine-tagged wild-type and its mutants (E280A, E287A, E297A, E429A, E520A, and E526A) of GnT-V luminal domain, as well as mini-GnT-V were expressed in COS-7 cells (RIKEN Cell Bank) and purified from culture media through Ni$^{2+}$-column. Enzymatic activity was measured as follows: 70 ng of purified GnT-V was mixed with various concentrations of a pyridylamine-labeled GlcNAc-terminated bian-tennary glycan (GnGnbi-PA) in 10 μl of a reaction buffer containing 125 mM MES (pH 6.25), 10 mM EDTA, 200 mM GlcNAc, 0.5% (v/v) Triton X-100, and 1 mg/ml bovine serum albumin. For kinetic analysis with donor substrate, the reaction was carried out in the presence of 200 μM acceptor substrate (GnGnbi-PA) and 20, 10, 5, 2.5, or 1.25 mM donor substrate (UDP-GlcNAc) for 1 h at 37 °C. For acceptor substrate, the reaction was carried out in the presence of 20 mM donor substrate and 40, 19.61, 8.39, 2.95, or 2.21 μM acceptor substrate for 15 min at 37 °C. For comparison between oligosaccharide-type substrate (GnGnbi-PA) and asparagine-type substrate GnGnbi-Asn-PNS, 10 μM acceptor substrate was used for enzymatic reaction for 1 h. After boiling for 5 min to stop the reaction, 40 μl of water was added, followed by centrifugation at 15,000 × g for 5 min. The supernatant was analyzed by reverse-phase HPLC (Prominence, Shimadzu) equipped with an ODS column (TSKgel ODS-80TM, TOSOH Bioscience). K$_m$ and V$_{max}$ values were cal-culated by nonlinear regression method using GraphPad Prism 7.

**Data availability.** Crystallographic data that support the findings of this study have been deposited in Protein Data Bank (PDB) with the accession codes of 5ZIB and 5ZIC. The other data that support the findings of this study are available from the corresponding author upon reasonable request.

## Table 2 Data collection and refinement statistics

| | Apo form (GnT-V luminal domain) | Ligand complex (Mini-GnT-V E297A) | Sulfur-SAD (GnT-V luminal domain) |
|---|---|---|---|
| **Data collection** | | | |
| Space group | P6$_5$22 | P2$_1$ | P6$_5$22 |
| Cell dimensions | | | |
| a, b, c (Å) | 97.7, 97.7, 268.9 | 70.4, 89.2, 92.2 | 97.7, 97.7, 270.4 |
| α, β, γ (°) | 90, 90, 120 | 90, 105.5, 90 | 90, 90, 120 |
| Wavelength | 0.9000 | 1.0000 | 2.7000 |
| Resolution (Å) | 48.1–1.90 (1.94–1.90) | 44.6–2.10 (2.15–2.10) | 48.8–2.72 (2.85–2.72) |
| R$_{sym}$ (%) | 21.8 (150.7) | 9.9 (129.3) | 20.1 (337.9) |
| I / σI | 9.5 (2.1) | 16.9 (1.7) | 37.1 (1.4) |
| Completeness (%) | 100 (100) | 98.5 (98.4) | 99.8 (98.4) |
| Redundancy | 14.5 (14.5) | 7.7 (7.8) | 95.4 (27.8) |
| **Refinement** | | | |
| Resolution (Å) | 48.1–1.90 | 44.6–2.10 | |
| No. reflections | 60,665 | 63,146 | |
| R$_{work}$/R$_{free}$ | 18.9/22.4 | 21.5/26.8 | |
| No. atoms | | | |
| Protein | 4711 | 7961 | |
| Ligand/ion | 14 | 74 | |
| Water | 294 | 218 | |
| B-factors | | | |
| Protein | 30.8 | 42.8 | |
| Ligand/ion | 36.7 | 42.3 | |
| Water | 29.8 | 34.1 | |
| R.m.s deviations | | | |
| Bond lengths (Å) | 0.007 | 0.003 | |
| Bond angles (°) | 0.875 | 0.664 | |

All datasets were obtained from single crystals
Values in parentheses are for highest-resolution shell

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

## Acknowledgements

We are grateful to Prof. Toshiyuki Shimizu for giving us the opportunity to undertake this research project. We are also grateful to Noriko Tanaka for secretarial assistance and Dr. Yusuke Yamada, Dr. Naohiro Matsugaki and Prof. Toshiya Senda (KEK) for collecting the native SAD dataset. This work was supported in part by Grant-in-Aid for Scientific Research Young scientist (B) (no. 15K18496) and Innovative Areas (no. 26110724, Deciphering sugar chain-based signals regulating integrative neuronal functions) to M.N., Scientific Research (C) (no. 17K07303 to M.N.; no. 17K07356 to Ya.K. and no. 25460054 to Y.Y.), and Leading Initiative for Excellent Young Researchers (LEADER) project to Ya.K. from the Ministry of Education, Culture, Sports, Science, and Technology (MEXT) of Japan. This work was also supported in part by the Platform Project for Supporting Drug Discovery and Life Science Research (Basis for Supporting Innovative Drug Discovery and Life Science Research (BINDS)) from Japan Agency for Medical Research and Development (AMED) under grant number JP17am0101075.

## Author contributions

M.N. directed the project and performed crystallographic experiments. Ya.K. performed DNA constructions and enzymatic assays. E.M. and J.T. carried out protein expression. Yu.K. contributed diffraction experiments. S.H. and Y.I. were responsible for chemical synthesis of inhibitor. N.T. and J.T. interpreted the data and commented on the manuscript. M.N. drafted the manuscript, and M.N., Ya.K., and Y.Y. wrote the manuscript.

## Additional information

**Competing interests:** The authors declare no competing interests.

