## [Peer Review File · Nature Communications]

Reviewers' comments:

Reviewer #1 (Remarks to the Author):

This paper describes the crystal-structure of accessible, catalytically active and purposefully mutated inactive, versions of GnT-V. The structures provide data on how the acceptor substrate binds to the enzyme surface, which in turn, provides a compelling model for why this glycosyltransferase cannot act after the action of GnT-III (a big deal in the glycobiology world) and for how acceptor protein surface structure might modulate accessibility of GnT-V, explaining why not all N-glycans can be substrates. Further, the authors are able to postulate (for the first time) an enzymatic mechanism for this GT18-family member and identify E297 as the key catalytic base.

Overall, the hard-won data are carefully described and appropriately discussed, with necessary caveats included. I found this to be a rather complete study.

Mike Ferguson, university of Dundee

Reviewer #2 (Remarks to the Author):

The enzyme MGATV/ GnT-V is one of the most important enzymes in human glycobiology and cancer. It catalyses the synthesis of beta 1-6 linked GlcNAc on cells, involved in forming a galactin lattice in cells and intimately linked to metastatic progression. In a landmark 2000 paper it was shown that *mgat5* ^{-/-} knockout mice show a marked reduction in the growth and metastasis of tumours whilst remaining, by other criteria, healthy.

The structure of MGATV has long been sought-after, both from a fundamental academic perspective and because it would provide a template for rational drug design on a major anti-cancer target. Many laboratories worldwide, including my own (sadly! I declare the conflict that we (only) have native data), have been seeking its 3-D structure.

The submission by Nagae and colleagues is therefore a massive and brilliant breakthrough and eminently suitable for publication Nature Communications (if not Nature itself). Not only does it provide the structure, and a ligand complex, it allows discussion of specificity, human variants, etc.

This is phenomenally good work.

My only concern, is that primarily the study reports the 3-D structure of MGATV in unliganded form and in complex with a bespoke bisubstrate inhibitor from a cunning variant. But I really struggled to see what the actual variant(s) was/were as the text seems highly contradictory at times.

Questions

I was a little confused what constructs were actually solved.

1. Was the "apo" structure simply T121-L741 ?
2. And then the complex obtained with the multi-mutant
 - a) E297A (later shown to be putative catalytic base?)
 - b) K330-R334 being replaced by 4 glycine residues. (as stated in methods...or was it a much bigger change as explained in the results?)

I think that could be better explained in the submission.

The nature of the loop change is also unclear.

"Second, the loop region (K330-R334) was exchanged to four consecutive glycine residues by PCR" That is a 4 residue substitution.

But later on it says the "in which N-terminal domain was deleted and the loop (K329-I345) was replaced with a short linker". That is (a) a removal of the first 213 residues and then (b) a 16 residue deletion, with "a short linker" which is undefined here This appears to be the same loop change as was differently described in the methods as a 4 residue substitution.

So I am left very confused by what construct was used for what crystals.

Minor issues

It is odd in the modern age to see "Km and Vmax values were calculated by double reciprocal Lineweaver-Burk plots". Surely linear regression software is available? People haven't used LB plots for Km analyses since the late 70s / early 80s because of their many flaws.

Without actually making a model I am not sure, but could you check that the GlcNAc-3 on Fig 1B is a D sugar. Easy to make a mistake with bond shading (i.e is C1-C2-C3-C4 really towards us, as opposed to C1, O5 C5 C4). As I said, I am not sure.

In my PDF, the blue shading on GlcNAc squares is not the blue used in the unified Glyco nomenclature (far too dark) and the green mannose isn't great either. I found that confusing.

Reviewer #3 (Remarks to the Author):

Manuscript No.: NCOMMS-18-11950-T

Title: Structure and mechanism of cancer-associated glycosyltransferase, N-acetylglucosaminyltransferase-V

Authors: Masamichi Nagae, Yasuhiko Kizuka Emiko Mihara, Yu Kitago, Shinya Hanashima, Yukishige Ito, Junichi Takagi, Naoyuki Taniguchi, and Yoshiki Yamaguchi

This manuscript describes the structure determination of a highly challenging glycoenzyme target, human MGAT5, which plays a critical role in the synthesis of highly branched complex N-glycan structures that are significantly upregulated in many types of cancer and possibly other diseases. The authors perform N-terminal truncation studies on MGAT5 to identify an effectively expressed and secreted forms of the enzyme. They then solve the crystal structure of one of the truncated enzyme forms and subsequently generate a structure of a more truncated enzyme in the presence of a chimeric substrate analog. Computational modeling studies were performed with alternative substrate structures to identify the steric constraints that provide acceptor specificity for the enzyme. This is particularly interesting based on the substrate specificity known for MGAT5.

The structural study is well-performed on this, an exceptionally high impact glycoenzyme, and it provides insights regarding the structural basis of the restricted substrate specificity for MGAT5. It also provides insights into the unusual nature of the glycosyltransferase structure that deviates significantly from more 'classical' GT-A fold, divalent cation-dependent GlcNAc transferases. The result is a novel paradigm for selective glycan extension by a metal-independent GT-B fold GlcNAc transferase.

The manuscript is generally well-written, but there are numerous places where additional clarification and editing are definitely required before the manuscript would be considered appropriate for publication. In addition, there are several major concerns that dampen enthusiasm for the manuscript including:

1) The authors take an interesting progression in their structural studies on MGAT5, by contrast to prior structural studies on GTs by other investigators. The authors initially perform a successful structure determination of MGAT5 as an apo-protein using a challenging intrinsic sulfur phasing approach. They then further truncate and mutate the protein and perform additional structural studies in the presence of a chimeric ligand (UDP-GlcNAc linked to GlcNAc-beta1,2-Man-alpha1,6-Man-alpha-octyl). In these latter studies, they were able to identify a bound complex for the GlcNAc-beta1,2-Man-alpha1,6-Man component, but not the UDP-GlcNAc or linker components. Thus, the authors were able to map the determinants for acceptor interactions, but not aspects of sugar-nucleotide donor binding site. A more 'standard' approach for GT structural studies would have been to progress from a UDP (and possibly UDP-GlcNAc) complex to a UDP + acceptor complex. It is not clear if the authors attempted to crystallize with UDP or acceptor alone. If so, then they should have commented on these results in the manuscript.

2) The point above is clearly relevant, because efforts were subsequently made to model the position of the UDP-GlcNAc donor based on the structure of a related bacterial GlcNAc transferase (BaBshA) that was previously determined in the presence of UDP (PDB 3MBO, Supplementary Fig. 3, ref 49). The problem is that the bacterial GT has a retaining catalytic mechanism of GlcNAc transfer, while MGAT5 is an inverting enzyme and likely positions the sugar nucleotide (or at least its sugar component) quite differently in the active site. If the authors were able to solve an MGAT5:UDP complex they would be able to test the similarity of the binding mode for the donor UDP component and validate the model for the position of the GlcNAc residue. The authors state that they modeled the position of the UDP-GlcNAc complex in MGAT5 based on the structure of the BaBshA:UDP complex (PDB 3MBO), but the approach for positioning the GlcNAc residue was, for some reason, not described. A more appropriate structure for modeling the position of the GlcNAc residue in BaBshA is PDB 5D00, where the structure of the BaBshA:UDP:GlcNAc-malate complex directly shows the position of the GlcNAc residue in this retaining GT. This latter structure should have been used in Supplementary Fig. 3.

3) The largest reason for all of the concern above is that the mechanism shown in Fig. 5 either is very poorly described and illustrated, or it makes absolutely no sense. The bound acceptor and the modeled sugar nucleotide donor do not position the C1 of the GlcNAc anywhere near the O6 of the Man nucleophile to carry out a direct displacement reaction. E297 is also nowhere near the Man O6 to act as a catalytic base. Fig 5 does not describe this quandary well at all, and the distance measurement that is shown in Fig 5 is not relevant to the catalytic mechanism; it is only the position of the missing linker region in the chimeric substrate analog. The authors must make a far better description of how they assembled the model in Fig 5 and why it does not seem to reflect an expected geometry for a direct displacement reaction by the enzyme. Without a better explanation for the mechanism it calls into question all aspects of the position of the donor in the active site and how the enzyme specifies the Man O6 as the nucleophile in the acceptor.

4) The authors provide an initial fragmentary logic for the choice of E297 as the potential catalytic base as a result of their mutagenesis and enzyme activity studies, but it would have made far more sense to make this determination after comparison with the BaBshA structure. Does the position of E297 make sense by comparison with the 5D00 structure? In this regard, E297 is not even shown in Supplementary Fig 3B (left panel), and it is probably the most important amino acid that should be shown in this figure. If E297 is not positioned appropriately, is there another putative catalytic base that makes more sense? It is also unclear why the authors used the E297A mutant for the generation of the mini-GnT-V construct. Were they expecting to reduce hydrolysis of the ligand? Was there any evidence that the chimeric ligand is cleaved by the wild type enzyme? The choice of this mutation for the acceptor structure complex clearly requires an explanation.

Other more minor concerns include:

1) Page 6 lower paragraph: The authors contrast the MGAT5 mechanism with MGAT1, but could

also include structures of MGAT2, C2GnT, POMGNT1 and others.

2) Page 7, upper paragraph: A more detailed description of the Dali search results is required. Were there other Dali hits? What were the Z-scores? Were they all in GT4? Why did they choose only BaBsshA for discussion? Did any of the other enzymes have an inverting catalytic mechanism? These are all questions to be addressed.

3) Page 7, middle paragraph: This description should be moved to after the description of the structure comparison with BaBshA so that it will make more sense in regard to the description of positions of the Glu residues relative to the donor ligand.

4) Page 7, lower paragraph: It is unclear, based on Fig 2, why the internal deletion (K329-I345) was performed for the mini-GnT-V construct and replaced by a linker. This region of the structure should be highlighted in Fig. 2A or 2C.

5) Page 8, bottom paragraph: Fig 3D and the discussion in this paragraph indicate that the transfer to the GlcNAc-beta1,2-Man-alpha1,3-Man structure is unfavorable based on steric reasons, but it appears that a different phi/psi rotamer of the alpha1,3Man linkage could make binding permissible. Was this tested?

6) Page 9, middle paragraph: The comparison of activities toward the two substrates shown in Fig 4D (data shown in Fig 4E) is not really a legitimate comparison, since the two compounds have very different linkers and tags which could account for the differences in activity.

7) Page 9, bottom paragraph: This entire paragraph and discussion needs to be revised based on the discussion above. I am not sure that Fig 5 makes any sense in regard to the catalytic mechanism, especially based on how the position of the UDP-GlcNAc was assembled in the figure. It is just as likely that E297 is not the catalytic base and the entire figure for the catalytic mechanism must be revised. The authors should really determine the structure of the MGAT5:UDP complex, at least to provide an initial basis for modeling-in the full UDP-GlcNAc structure (or at least state that they tried and failed).

8) Page 12, top paragraph: The discussion of the positions of residues in the chimeric ligand (S6 and C8) are confusing based on the structure of the ligand shown in Fig 1B. There should be labeling of these positions in this figure panel, and it should refer to the panel in the discussion so that a reader knows what is being described.

9) Page 12, bottom paragraph: I am not sure that I would consider cadherins to be 'giant glycoproteins'. Possibly a different terminology should be used!

10) Fig. 2C: the side chains of E287 and E290 are missing in the figure.

11) Supplementary Fig 3B, left panel: show the side chain of E297 in the figure.

12) Supplementary Fig 5B: There is poor color contrast between the surface representation of the protein and the surface of the spheres for the ligand. This should be improved.

13) Supplementary Fig 8: There is poor color contrast between the color of the cartoon helices and the sphere representation of L189. This should be improved.

In sum, this is clearly a high-impact manuscript describing the structure and structural mechanism of a glycosyltransferase of critical importance. There are significant questions that must be addressed, however.

Point-by-point reply

Reviewers' comments:

Reviewer #1

Comment 1:

This paper describes the crystal-structure of accessible, catalytically active and purposefully mutated inactive, versions of GnT-V. The structures provide data on how the acceptor substrate binds to the enzyme surface, which in turn, provides a compelling model for why this glycosyltransferase cannot act after the action of GnT-III (a big deal in the glycobiology world) and for how acceptor protein surface structure might modulate accessibility of GnT-V, explaining why not all N-glycans can be substrates. Further, the authors are able to postulate (for the first time) an enzymatic mechanism for this GT18-family member and identify E297 as the key catalytic base.

Overall, the hard-won data are carefully described and appropriately discussed, with necessary caveats included. I found this to be a rather complete study.

Response:

We really appreciate the reviewer's positive comments on our manuscript. We believe that our paper will provide insights into how glycosyltransferases recognize their substrates.

Reviewer

#2

Comment

1:

The enzyme MGATV/ GnT-V is one of the most important enzymes in human glycobiology and cancer. It catalyzes the synthesis of beta 1-6 linked GlcNAc on cells, involved in forming a galectin lattice in cells and intimately linked to metastatic progression. In a landmark 2000 paper it was shown that *mgat5* ^{-/-} knockout mice show a marked reduction in the growth and metastasis of tumors whilst remaining, by other criteria, healthy.

The structure of MGATV has long been sought-after, both from a fundamental academic prospective and because it would provide a template for rational drug design on a major anti-cancer target. Many laboratories worldwide, including my own (sadly! I declare the conflict that we (only) have native data), have been seeking its 3-D structure.

The submission by Nagae and colleagues is therefore a massive and brilliant breakthrough and eminently suitable for publication Nature Communications (if not Nature itself). Not only does it provide the structure, and a ligand complex, it allows discussion of specificity, human variants, etc.

Response:

Thank you very much for your positive comments. According to your questions and suggestions, we have revised the manuscript as below. We hope the revised paper is now acceptable.

Comment 2:

This is phenomenally good work. My only concern, is that primarily the study reports the 3-D structure of MGATV in unliganded form and in complex with a bespoke bisubstrate inhibitor from a cunning variant. But I really struggled to see what the actual variant(s) was/were as the text seems highly contradictory at times.

Questions.

I was a little confused what constructs were actually solved.

1. Was the “apo” structure simply T121-L741?

Response

The construct used for apo form structure was T121-L741. We clarified this point as “We determined the crystal structure of GnT-V luminal domain (T121-L741) in apo form” in line 17 of page 6.

2. And then the complex obtained with the multi-mutant

a) E297A (later shown to be putative catalytic base?)

b) K330-R334 being replaced by 4 glycine residues. (as stated in methods...or was it a much bigger change as explained in the results?)

I think that could be better explained in the submission.

Response

The construct used for ligand complex structure was multi-mutant named as mini-GnT-V E297A which lacks both N-terminal domain and a long unstructured loop (K330-R344) and has point mutation E297A. The deletion of loop region (K330-R344) means that K329 is connected to I345 via a short glycine linker. We are very sorry for the confusion. We miswrote the Method and should have written “the loop region (K330-R344)”, instead of “the loop region (K330-R334)” in line 2 of page 4. Now, we have corrected the text and clearly described in both Method and Results that the N-terminal region (T121-H212) was deleted and the loop region (K330-R344) was replaced as GGGG in the mini-GnT-V construct. To clarify this point, we made a new figure illustrating the differences of the two constructs used for crystallization in apo and ligand-bound forms as Figure 3B. Since mini-GnT-V has enough catalytic activity (Figure 1D), the bisubstrate inhibitor may be degraded during crystal

formation. Therefore, we further introduced the E297A mutation for stable complex formation. We have now highlighted the position of E297A and labeled Figure 3A appropriately.

The nature of the loop change is also unclear. “Second, the loop region (K330-R334) was exchanged to four consecutive glycine residues by PCR”. That is a 4 residue substitution. But later on it says the “in which N-terminal domain was deleted and the loop (K329-I345) was replaced with a short linker”. That is (a) a removal of the first 213 residues and then (b) a 16 residue deletion, with “a short linker” which is undefined here. This appears to be the same loop change as was differently described in the methods as a 4 residue substitution. So I am left very confused by what construct was used for what crystals.

Response:

As explained above, we replaced K330-R344 with four consecutive glycine residues as well as the deletion of N-terminal domain.

Minor issues

Comment 3:

It is odd in the modern age to see “Km and Vmax values were calculated by double reciprocal Lineweaver-Burk plots”. Surely linear regression software is available? People haven’t used LB plots for Km analyses since the late 70s/ early 80s because of their many flaws.

Response:

Thank you for your suggestion. Kinetic parameters are now obtained by curve fitting. The graph and table in Figure 2D have been changed appropriately. We added the text in line 3 of page 6 as follows: “ K_m and V_{max} values were calculated by nonlinear regression method”.

Comment 4:

Without actually making a model I am not sure, but could you check that the GlcNAc-3 on Fig 1B is a D sugar. Easy to make a mistake with bond shading (i.e is C1-C2-C3-C4 really towards us, as opposed to C1, O5 C5 C4). As I said, I am not sure.

Response:

We carefully checked the structure, and by making a model as shown below, we confirmed that the chirality of GlcNAc-3 is correct (Fig. R3).

Fig. R3 Chemical structures of GlcNAc viewed from two different angles.

Comment 5:

In my PDF, the blue shading on GlcNAc squares is not the blue used in the unified Glyco nomenclature (far too dark) and the green mannose isn't great either. I found that confusing.

Response:

Thank you very much for pointing this out. We corrected the colors of these symbols throughout the manuscript to conform to the Symbol Nomenclature for Glycans.

Reviewer

#3

This manuscript describes the structure determination of a highly challenging glycoenzyme target, human MGAT5, which plays a critical role in the synthesis of highly branched complex N-glycan structures that are significantly upregulated in many types of cancer and possibly other diseases. The authors perform N-terminal truncation studies on MGAT5 to identify an effectively expressed and secreted forms of the enzyme. They then solve the crystal structure of one of the truncated enzyme forms and subsequently generate a structure of a more truncated enzyme in the presence of a chimeric

substrate analog. Computational modeling studies were performed with alternative substrate structures to identify the steric constraints that provide acceptor specificity for the enzyme. This is particularly interesting based on the substrate specificity known for MGAT5.

The structural study is well-performed on this, an exceptionally high impact glycoenzyme, and it provides insights regarding the structural basis of the restricted substrate specificity for MGAT5. It also provides insights into the unusual nature of the glycosyltransferase structure that deviates significantly from more 'classical' GT-A fold, divalent cation-dependent GlcNAc transferases. The result is a novel paradigm for selective glycan extension by a metal-independent GT-B fold GlcNAc transferase.

The manuscript is generally well-written, but there are numerous places where additional clarification and editing are definitely required before the manuscript would be considered appropriate for publication. In addition, there are several major concerns that dampen enthusiasm for the manuscript including:

Comment 1:

The authors take an interesting progression in their structural studies on MGAT5, by contrast to prior structural studies on GTs by other investigators. The authors initially perform a successful structure determination of MGAT5 as an apo-protein using a challenging intrinsic sulfur phasing approach. They then further truncate and mutate the protein and perform additional structural studies in the presence of a chimeric ligand (UDP-GlcNAc linked to GlcNAc-beta1,2-Man-alpha1,6-Man-alpha-octyl). In these latter studies, they were able to identify a bound complex for the GlcNAc-beta1,2-Man-alpha1,6-Man component, but not the UDP-GlcNAc or linker components. Thus, the authors were able to map the determinants for acceptor interactions, but not aspects of sugar-nucleotide donor binding site. A more 'standard' approach for GT structural studies would have been to progress from a UDP (and possibly UDP-GlcNAc) complex to a UDP + acceptor complex. It is not clear if the authors attempted to crystallize with UDP or acceptor alone. If so, then they should have commented on these results in the manuscript.

Response:

We fully agree with the reviewer's comment that starting with a UDP complex would be first choice to determine the position of donor substrate for GnT-V. Actually, we did try it many times, but all unfortunately failed. We initially tried to crystallize wild type GnT-V luminal domain in complex with UDP or UDP-GlcNAc by co-crystallization as well as soaking methods, but electron density of UDP moiety was not observed at all. Next, we introduced the single point mutation at E297, E520 or E526 in GnT-V luminal domain and tried to crystallize these mutants with various ligands in the same way. However, we again could not obtain any clear electron density of either UDP or UDP-GlcNAc. Therefore, we made a new construct, mini-GnT-V, and co-crystallized it with the bisubstrate inhibitor as described in the paper. We also tried to crystallize mini-GnT-V in complex with UDP or UDP-GlcNAc, but again we were not able to observe clear electron density of UDP and UDP-GlcNAc. Therefore, the only course open to us was to speculate the position of donor substrate by structural comparison with the BaBshA complex.

Because of your criticism, we added a brief summary of our trials as follows, “To obtain complex structures with donor and acceptor substrates, we tried to crystallize wild type, E297A, E520A or E526A GnT-V luminal domain in complex with donor UDP-GlcNAc or UDP and/or acceptor *N*-glycans by co-crystallization as well as soaking methods. Among the three mutants, E297A was the only one successfully crystallized even in the presence of ligands, although no clear electron density of substrates was observed. As a result of these trials, we decided to change to the construct.” (line 1-5 in page 8).

Comment 2:

The point above is clearly relevant, because efforts were subsequently made to model the position of the UDP-GlcNAc donor based on the structure of a related bacterial GlcNAc transferase (BaBshA) that was previously determined in the presence of UDP (PDB 3MBO, Supplementary Fig. 3, ref 49). The problem is that the bacterial GT has a retaining catalytic mechanism of GlcNAc transfer, while MGAT5 is an inverting enzyme and likely positions the sugar nucleotide (or at least its sugar component) quite differently in the active site. If the authors were able to solve an MGAT5:UDP complex they would be able to test the similarity of the binding mode for the donor UDP component and validate the model for the position of the GlcNAc residue. The authors state that they modeled the position of the UDP-GlcNAc complex in MGAT5 based on the structure of the BaBshA:UDP complex (PDB 3MBO), but the approach for positioning the GlcNAc residue was, for some reason, not described. A more appropriate structure for modeling the position of the GlcNAc residue in BaBshA is PDB 5D00, where the structure of the BaBshA:UDP:GlcNAc-malate complex directly shows the position of the GlcNAc residue in this retaining GT. This latter structure should have been used in Supplementary Fig. 3.

Response:

Thank you for your constructive comment. First, as described above, crystallization of UDP complex was not successful.

As the reviewer suggested, we included BaBshA:UMP:GlcNAc-malate complex (PDB code: 5D00) as well as BaBshA:UDP complex (PDB code: 3MBO) in Suppl. Figure 3. We also added the six glutamate residues tested in the mutational experiments in the same figure. Although BaBshA is a retaining enzyme as pointed out, structural superposition showed that the positions of ribose in both UDP and UMP:GlcNAc-malate complexes are close to E526, and an alanine mutant of this residue completely abolished activity of GnT-V. The position of donor substrate in BaBshA does appear to be close to that in GnT-V.

In the first version of our manuscript, the ternary complex model was built by simply superposing an ideal UDP-GlcNAc structure deposited in monomer library of CCP4 onto UDP moiety deduced from the UDP complex. However, the reviewer has made us relook at the model, and the figure and paragraph are completely revised as described in our responses below.

Comment 3:

The largest reason for all of the concern above is that the mechanism shown in Fig. 5 either is very poorly described and illustrated, or it makes absolutely no sense. The bound acceptor and the modeled sugar nucleotide donor do not position the C1 of the GlcNAc anywhere near the O6 of the Man nucleophile to carry out a direct displacement reaction. E297 is also nowhere near the Man O6 to act as a catalytic base. Fig 5 does not describe this quandary well at all, and the distance measurement that is shown in Fig 5 is not relevant to the catalytic mechanism; it is only the position of the missing linker region in the chimeric substrate analog. The authors must make a far better description of how they assembled the model in Fig 5 and why it does not seem to reflect an expected geometry for a direct displacement reaction by the enzyme. Without a better explanation for the mechanism it calls into question all aspects of the position of the donor in the active site and how the enzyme specifies the Man O6 as the nucleophile in the acceptor.

Response:

We responded to this comment collectively with the next comment.

Comment 4:

The authors provide an initial fragmentary logic for the choice of E297 as the potential catalytic base as a result of their mutagenesis and enzyme activity studies, but it would have made far more sense to make this determination after comparison with the BaBshA structure. Does the position of E297 make sense by comparison with the 5D00 structure? In this regard, E297 is not even shown in Supplementary Fig 3B (left panel), and it is probably the most important amino acid that should be shown in this figure. If E297 is not positioned appropriately, is there another putative catalytic base that makes more sense? It is also unclear why the authors used the E297A mutant for the generation of the mini-GnT-V construct. Were they expecting to reduce hydrolysis of the ligand? Was there any evidence that the chimeric ligand is cleaved by the wild type enzyme? The choice of this mutation for the acceptor structure complex clearly requires an explanation.

Response to Comments 3 and 4:

We agree that great care needs to be taken in making a ternary complex model and in deducing the catalytic residue. We have carefully redrawn Figure 5 and substantially rewrote the corresponding parts in the text.

First, according to the reviewer's suggestion, we compared the structures of not only BaBshA:UDP complex but also BaBshA:UMP:GlcNAc-malate complex (PDB code: 5D00) with our GnT-V apo form (Suppl. Figure 3) or GnT-V:ligand complex (Figure 5A and 5B). UDP and UMP moieties in the two BaBshA complexes are located almost at the same place (Figure 5A and 5B) and both fit well in the GnT-V:ligand complex. In addition, mutation of E526, which is close to the position of UDP deduced from the BaBshA complex, almost completely abolished enzyme activity, suggesting that the position of UDP in GnT-V can be inferred from the BaBshA complex. In contrast, the GlcNAc

moiety of BaBshA:UMP:GlcNAc-malate complex, when simply imposed onto GnT-V, causes steric clash with the loop (P518-P522) of GnT-V (Figure 5B), indicating that the position of GlcNAc in BaBshA:UMP:GlcNAc-malate complex cannot be used for making a model without substantial changes. Thus, we first fixed the position of the UDP moiety based on BaBshA:UDP complex and superposed UDP-GlcNAc extracted from GnT-I-UDP-GlcNAc complex (PDB code: 1FOA) onto the position of UDP moiety by least square fitting (Figure 5C).

Even in this revised model, the distance between Man O6 and GlcNAc C1 is 6.0Å, which is a rather distant for direct displacement, compared with GnT-I and II in which the catalytic residue is commonly located more closely to the attacked oxygen (Suppl. Figure 7A). This may be derived from the limitations of model building using multiple structural superposition. However, in this model, as well as in our previous model, E297 is the only polar residue located within 5 Å of Man O6 (Figure 5D) - there is no other candidate for catalytic base in the vicinity. As repeatedly mentioned, mutation of E297 completely abolishes activity. This residue must be the catalytic base. Importantly E297 resides at the edge of a loop (G282-G296), which is missing both in the apo form and in molecule A of the ligand-bound form and thus likely to be highly flexible, and therefore the 6 Å or so distance may be shortened for catalysis. In fact, E297 would make hydrogen bond with OH6 of Mannose with a slight rotation of OH6 (Figure 5C). Based on these considerations, we have substantially changed the whole paragraph as follows, “Crystallographic and biochemical studies provide a rational catalytic reaction mechanism of GnT-V. Structural superposition of the unliganded form of wild type GnT-V and the ligand complex of mini-GnT-V E297A shows that E297 is the only acidic residue within 5Å of OH6 (S6) of Man-2 (Figure 5A). This strongly suggests that E297 functions as a catalytic base, consistent with the mutational analysis (Figure 2D). In comparison with the *BaBshA* structures, the UDP (or UMP) moiety of *BaBshA*-ligand complexes fit into the groove of GnT-V without apparent clash, but the product GlcNAc-malate clashes with the loop (P518-P522) of GnT-V (Figure 5B). Therefore, we built in donor substrate, UDP-GlcNAc, to the GnT-V-ligand structure, based on the superposition of UDP-GlcNAc extracted from GnT-I donor complex (PDB code: 1FOA⁴⁵) onto UDP moiety position by least square fitting (Figure 5C). In this GnT-V ternary complex model, the likely catalytic base E297 is located within hydrogen bond distance of OH6 of Man-2 (3.2Å), but the distance between OH6 of Man-2 and C1 atom of UDP-GlcNAc is 6.0Å, requiring a positional shift for efficient catalysis. In other GlcNAc transferases with GT-A folds, the catalytic base residues is located closer to the hydroxyl group of acceptor substrate (Suppl. Figure 7). In the GnT-V-ligand structure, E297 is located at the base of a highly mobile loop (G282-G296), and it is conceivable that this loop, and the putative catalytic residue, moves to effect catalysis (Figure 5A). Apart from E297, there are five other glutamates in the binding cleft, and all six were tested in mutational experiments (Figure 2D). Glutamates E429, E520 and E526 are close to the putative donor binding site; E526 is near the ribose moiety, and so is E429, though slightly further away in the present model, consistent with mutant E429A retaining enzymatic activity, while, E520 lies close to GlcNAc and E520A severely impairs activity. This latter residue is slightly buried and forms a tight salt bridge with R558 and may also play a role in donor interaction. The other two glutamates, E280 and E287, are more apart from donor and acceptor substrate, and mutants E280A and E287A retain enzymatic activity, indicating only supportive roles (Figure 2D and 5D).” (page 10, line 6-28).

Regarding E297A mutation, we tested E297A, E520A and E526A, as well as wild-type of GnT-V luminal domain for making a ligand complex during our trials. Although we have no direct evidence

to show the wild-type mini-GnT-V degrades the chimeric ligand, as the reviewer assumed, we expected that mutation at these critical residues could make ligands stable by blocking hydrolysis during crystal formation. Among the mutants, E297A was only one successfully crystallized with ligands, although their detailed positions cannot be assigned. From this result, we concluded that the E297A mutation was effective to get a ligand complex. Then we switched the construct to mini-GnT-V with E297A mutant from the first trial. We have briefly described the reason why we used E297A mutant, “Among the three mutants, E297A was the only one which allowed successful crystallization even in the presence of ligands, although no clear electron density of the substrates was observed.” in the text (page 8, line 2-4).

Other more minor concerns include:

Comment 5:

Page 6 lower paragraph: The authors contrast the MGAT5 mechanism with MGAT1, but could also include structures of MGAT2, C2GnT, POMGNT1 and others.

Response:

As suggested, we have now made close-up views of the catalytic centers of MGAT1 (GnT-I), MGAT2 (GnT-II), C2GnT and POMGnT1 in Suppl. Figure 7, and we revised the text in line 23-25 of page 6, “and is in marked contrast to other mammalian GlcNAc transferases such as GnT-I (GT13)⁴⁵, GnT-II (GT16)⁴⁶, POMGnT1⁴⁷ and C2GnT-L⁴⁸ which are classified into GT-A folds” and in line 17-19 of page 10 “In other GlcNAc transferases with GT-A folds, the catalytic base residues are commonly located close to the hydroxyl group of acceptor substrate (Suppl. Figure 7).”.

Comment 6:

Page 7, upper paragraph: A more detailed description of the Dali search results is required. Were there other Dali hits? What were the Z-scores? Were they all in GT4? Why did they choose only BaBsshA for discussion? Did any of the other enzymes have an inverting catalytic mechanism? These are all questions to be addressed.

Response:

In our DALI search, we examined all top 20 structures belonging to the GT4 family. We added details of the DALI search results as well as Z-score in line 4-9 in Page 7 as follows “A DALI search⁵¹ shows GnT-V has structural similarities to bacterial glycosyltransferases in the GT4 family such as first mannosyl transferase Wbaz-1 (PDB code: 2F9F, Z-score = 12.4), *BaBshA*, also referred to as *BaGT4*_{BA1558} (PDB code: 2JJM, Z-score = 12.0) and phosphatidylinositol mannosyltransferase (PDB code: 2GEJ, Z-score = 11.8). Although all these members are retaining enzymes, *BaBshA* also uses UDP-GlcNAc as donor substrate, as in the case of inverting GnT-V (Suppl. Figure 3A)”.

Comment 7:

Page 7, middle paragraph: This description should be moved to after the description of the structure comparison with BaBshA so that it will make more sense in regard to the description of positions of the Glu residues relative to the donor ligand.

Response:

We agree. Accordingly, we have added several sentences for structural comparison between GnT-V and BaBshA:UDP complex before the description about the mutants, “The position of E526 in GnT-V coincides well with that of E290 (E291) in the *BaBshA*-UDP complex (Suppl. Figure 3B). This glutamate of *BaBshA* directly interacts with the ribose moiety of the donor substrate. E520 is also located close, but this residue is slightly buried and forms a tight salt bridge with R558. In addition, several residues of *BaBshA* which interact with UDP-GlcNAc such as N206, K211 and E282 are not conserved in GnT-V. The lack of polar residues may be a reason for the weak donor binding of GnT-V, compared with other human GnTs^{53,54} (Suppl. Figure 1)” (page 7, line 10-16).

Comment 8:

Page 7, lower paragraph: It is unclear, based on Fig 2, why the internal deletion (K329-I345) was performed for the mini-GnT-V construct and replaced by a linker. This region of the structure should be highlighted in Fig. 2A or 2C.

Response:

We thought that deletion of only the N-terminal domain would leave the long unstructured loops (K330-S336 and T340-R344) and free cysteine (C338), which could be an obstacle for crystallization. Thus, we decided to replace the loop with four glycine residues in conjunction with the N-terminal domain deletion. As shown in the newly added Figure 3B, the N-terminal domain is linked to Middle domain 1 via three missing loops (E208-D211, K330-S336, and T340-R344), implying the highly mobile nature of this domain. Furthermore, an inter-regional disulfide bond was formed between C338 (in the loop) and C172 (in N-terminal domain).

We added the labels of three missing loops (E208-D211, K330-S336, and T340-R344) which connect between N-terminal domain and Middle domain 1 in Fig. 2A. We also added the following sentence about the missing loop “And, the N-terminal domain is linked to Middle domain 1 via three missing loop regions, signifying the highly mobile nature of this domain (Figure 2A)” in line 31-32 of page 6. In addition, we added the following sentence to clarify the reason for loop replacement “Also, the long unstructured loop (K330-R344), involved in an interregional disulfide bond (C172-C338), was replaced with a short linker of four glycine residues to reduce the structural heterogeneity (Figure 1C)” in line 7-10 of page 8.

Comment 9:

Page 8, bottom paragraph: Fig 3D and the discussion in this paragraph indicate that the transfer to the GlcNAc-beta1,2-Man-alpha1,3-Man structure is unfavorable based on steric reasons, but it appears that a different phi/psi rotamer of the alpha1,3Man linkage could make binding permissible. Was this tested?

Response:

If we test only the trisaccharide moiety, we do find the permissible rotamers. However, even in that case, the chitobiose part causes a serious clash with GnT-V. We built the new docking model and added it in Suppl. Figure 5A. We also added the sentence in line 9-11 of page 9, “Besides, even when we chose a rotamer which can fit into the active site, the chitobiose moiety sterically clashes with GnT-V (Suppl. Figure 5A)”.

Comment 10:

Page 9, middle paragraph: The comparison of activities toward the two substrates shown in Fig 4D (data shown in Fig 4E) is not really a legitimate comparison, since the two compounds have very different linkers and tags which could account for the differences in activity.

Response:

As the reviewer pointed out, in addition to the presence or absence of Asn, the two substrates are also different in linker length. Thus, we cannot exclude the possibility that the linker could affect the binding. However, at least our data suggest that a glycan core or peptide part of *N*-glycan substrate may impact on glycosyltransferase activity, which even acts on a peripheral part of *N*-glycan. We revised and toned down the text in line 1-3 of page 10, “This suggests that the core region of *N*-glycan does interact with GnT-V and the contacts contribute to substrate specificity, although it is also possible that the attached linker chain may modify the interaction”.

Comment 11:

Page 9, bottom paragraph: This entire paragraph and discussion needs to be revised based on the discussion above. I am not sure that Fig 5 makes any sense in regard to the catalytic mechanism, especially based on how the position of the UDP-GlcNAc was assembled in the figure. It is just as likely that E297 is not the catalytic base and the entire figure for the catalytic mechanism must be revised. The authors should really determine the structure of the MGAT5:UDP complex, at least to provide an initial basis for modeling-in the full UDP-GlcNAc structure (or at least state that they tried and failed).

Response:

As described above, our repeated trials for making a donor complex were all unsuccessful except the mini-GnT-V E297A data shown in the paper. We carefully revised Figure 5 and rewrote the paragraph in page 9-10 as described in the response to comments 3 and 4.

Comment 12:

Page 12, top paragraph: The discussion of the positions of residues in the chimeric ligand (S6 and C8) are confusing based on the structure of the ligand shown in Fig 1B. There should be labeling of these positions in this figure panel, and it should refer to the panel in the discussion so that a reader knows what is being described.

Response:

We put the labels of the two atoms in Fig 1B and clearly refer to the panel in page 12, line 27, thank you. Consistent with the revision of Figure 5, we revised the text of this part and now S6 and C8 were removed as follows, “Even so, it is possible that the donor moiety does not properly fit the cavity due to the linker length still not being long enough” in page 12, line 20-21.

Comment 13:

Page 12, bottom paragraph: I am not sure that I would consider cadherins to be ‘giant glycoproteins’. Possibly a different terminology should be used!

Response:

We replaced ‘giant’ to ‘multidomain’ in line 6 of page 13.

Comment 14:

Fig. 2C: the side chains of E287 and E290 are missing in the figure.

Response:

E280 and E287 are in the missing loop in apo form structure. We emphasized this point in line 24-25 of page 7 as follows “Among these residues, E280 and E287 are located in the missing loop (bottom line in Figure 2C).”.

Comment 15:

Supplementary Fig 3B, left panel: show the side chain of E297 in the figure.

Response:

We revised Suppl. Figure 3 and added the side chains of E297 as well as other glutamates.

Comment 16:

Supplementary Fig 5B: There is poor color contrast between the surface representation of the protein and the surface of the spheres for the ligand. This should be improved.

Response:

Thank you for your suggestion. We omitted semi-transparent sphere model and redrew the zoomed figure for clarity.

Comment 17:

Supplementary Fig 8: There is poor color contrast between the color of the cartoon helices and the sphere representation of L189. This should be improved.

Response:

We improved the figure, thank you.

Comment 18:

In sum, this is clearly a high-impact manuscript describing the structure and structural mechanism of a glycosyltransferase of critical importance. There are significant questions that must be addressed, however.

Response:

We really appreciate your thoughtful comments and very pleased to learn that our paper is “clearly a high-impact manuscript”. Actually, since we got the first crystal of an apo-form, we have been really struggling to get a donor (UDP) complex for over 2 years, with repeated failures except for the data shown in the paper. According to your comments, we carefully revised the manuscript, particularly regarding Fig. 5. We hope that the new version is acceptable.

REVIEWERS' COMMENTS:

Reviewer #3 (Remarks to the Author):

The authors have made a genuine effort to address our concerns and questions. We believe the manuscript is much stronger than when first submitted and reflects much work and dedication on the part of the authors.